# GraIP: A Benchmarking Framework For Neural Graph Inverse Problems

## Abstract

A wide range of graph learning tasks—such as structure discovery, temporal graph analysis, and combinatorial optimization—focus on inferring graph structures from data, rather than making predictions on given graphs. However, the respective methods to solve such problems are often developed in an isolated, task-specific manner and thus lack a unifying theoretical foundation. Here, we provide a stepping stone towards the formation of such a foundation and further development by introducing the *Neural Graph Inverse Problem* (GraIP) conceptual framework, which formalizes and reframes a broad class of graph learning tasks as inverse problems. Unlike discriminative approaches that directly predict target variables from given graph inputs, the GraIP paradigm addresses inverse problems, i.e., it relies on observational data and aims to recover the underlying graph structure by reversing the forward process—such as message passing or network dynamics—that produced the observed outputs. We demonstrate the versatility of GraIP across various graph learning tasks, including rewiring, causal discovery, and neural relational inference. We also propose benchmark datasets and metrics for each GraIP domain considered, and characterize and empirically evaluate existing baseline methods used to solve them. Overall, our unifying perspective bridges seemingly disparate applications and provides a principled approach to structural learning in constrained and combinatorial settings while encouraging cross-pollination of existing methods across graph inverse problems.

## 1 Introduction

In graph machine learning, numerous challenges—including structural optimization, causal discovery, and gene regulatory network reconstruction—focus on estimating underlying graph structures from observations, rather than performing inference on relational data. While recent graph-learning methods, e.g., *message-passing graph neural networks* (Gilmer et al., 2017; Scarselli et al., 2009) (MPNNs) have achieved impressive results on such individual graph problems, e.g., (heuristically) solving graph-based combinatorial optimization problems (Karalias & Loukas, 2020; Wenkel et al., 2024) or network inference tasks (Bhaskar et al., 2024), these approaches are often developed in isolation, tailored to specific tasks, and lack a unifying formalism. As a notable example, existing work on graph rewiring (Qian et al., 2023; 2024; Qiu et al., 2022) and graph structure learning (Fatemi et al., 2023) reveals that these domains, although typically pursued in isolation, are fundamentally concerned with the same problem, namely modifying or inferring graph structure from data. Both settings face nearly identical methodological challenges as well. That is, their separation largely reflects the absence of a unifying framework, rather than any principled distinction beyond their respective downstream objectives.

On the other hand, *inverse problems* arise as a common formulation spanning many domains in applied mathematics and engineering, where the goal is to infer the underlying causal factors that give rise to indirect and typically noisy observational data (Daras et al., 2024; Kirsch et al., 2011). Inverse problems have a rich history across fields such as signal processing, system identification, computer vision, and astronomy, where data-driven, machine learning-based methods now form a major class of approaches for tackling them (Daras et al., 2024; Kamyab et al., 2022; Zheng et al., 2025), in contrast to the earlier dominance of physics-driven analytical methods (Kirsch et al., 2011).

While it may initially appear that an extension to relational data domains, such as graph learning, is only natural, inverse problems on graphs seem to be largely overlooked in the relevant literature. We therefore

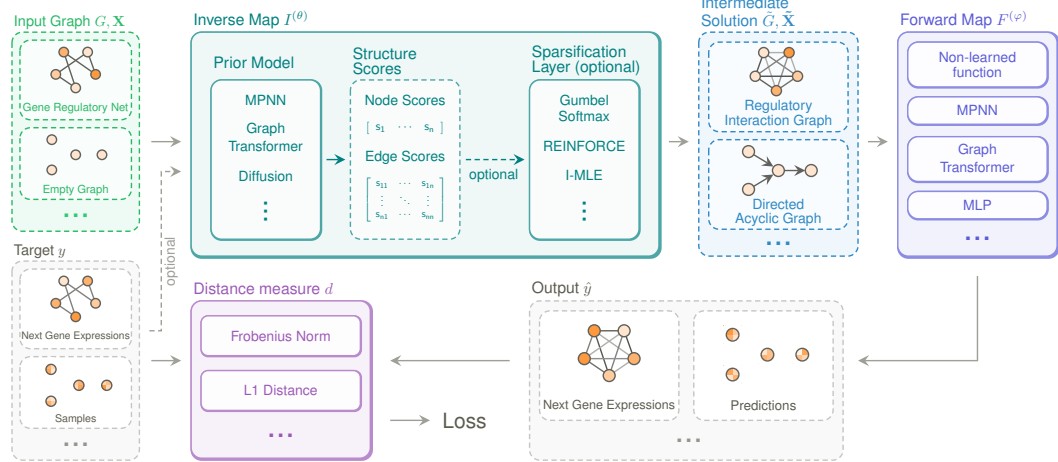

Figure 1: **Overview of the GraIP framework.** The input graph $G$ with optional node features $\boldsymbol{X}$ and target $y$ is fed into the inverse map $I^{(\boldsymbol{\theta})}$. This produces an intermediate solution graph $\tilde{G}$ with optional node features $\tilde{\boldsymbol{X}}$. The forward map $F^{(\boldsymbol{\varphi})}$ uses the intermediate solution to produce output $\hat{y}$, which is compared with $y$ using distance measure $d$ to compute the loss. The specific instantiation of each component depends on the domain. We show examples based on gene regulatory network inference and causal discovery.

draw inspiration from the above characterization of inverse problems and demonstrate that they naturally extend to a varied subset of graph learning problems.

We thus introduce the *Neural Graph Inverse Problem* (GraIP) benchmarking framework. This comprehensive formulation unifies a wide range of graph learning problems under a single umbrella by framing them as *inverse problems*. In the GraIP framework, we consider how a given forward process——representing, for example, the propagation of signals over a network (Graber & Schwing, 2020) or the dynamics of biological interactions (Bhaskar et al., 2024)—can be inverted to recover the underlying graph structures. This perspective bridges diverse applications, including causal inference, combinatorial optimization, and regulatory network reconstruction, by exposing their shared intrinsic components. In doing so, we provide a principled foundation for developing algorithms that are both comparable across domains and capable of addressing common challenges such as constraint satisfaction, non-identifiability, and differentiation through discrete combinatorial choices. We further provide baseline empirical results, establishing a basis for the systematic evaluation of methods within the GraIP framework. We present an overview of our framework in Figure 1.

**Present work** To the best of our knowledge, the inverse problem perspective has not yet been systematically applied to graph-structured data. In this work, we take a first step in this direction with the GraIP framework and provide a unified lens on solving diverse graph learning tasks as inverse problems. We envision GraIP as a stepping stone for future developments that can leverage novel ideas from the field of inverse problems to advance graph learning. Our key contributions are as follows.

1. We derive the GraIP framework, unifying a wide range of graph learning tasks, including causal discovery, structure learning, and dynamic graph inference, under the lens of inverse problems.

2. We instantiate our framework on diverse inverse graph problems and demonstrate how baseline methods that incorporate established graph learning tools, including MPNNs, graph transformers, and differentiable sampling, fit into GraIP.

3. We provide practical insights from our implementation of various GraIPs, demonstrate how these problems can be addressed within a unified pipeline, and discuss the current challenges in creating synergy across different domains.

*By framing many diverse graph learning problems as inverse tasks, our work provides a principled and versatile framework for tackling various challenges in graph-based machine learning.*

## 2 BACKGROUND

Here, we review related work and overview graph learning tasks relevant to our framework. Additional background and notation are provided in Appendix A.

### 2.1 INVERSE PROBLEMS

Inverse problems are a fundamental challenge across many domains in the natural sciences and engineering, and they concern inferring unknown causes or system parameters from noisy observational data. Prevalent examples include imaging problems like denoising and hyperspectral unmixing (Ongie et al., 2020), parameter estimation problems (Aster et al., 2012), compressed sensing for MRI (Lustig et al., 2008), and black hole imaging with radio telescope arrays (Zheng et al., 2025). Inverse problems are also relevant to systems governed by partial differential equations, like fluid dynamics, where the inversion process aims to recover the initial conditions based on observed flow measurements (Zhang et al., 2020).

In inverse problems, we have observations $\boldsymbol{y}$ derived from some latent source $\boldsymbol{z}$ via a *forward map* $F$. The inverse problem is to find an *inverse map* $I$ to infer the latent source $\boldsymbol{z}$ from observed $\boldsymbol{y}$, i.e.,

$$\boldsymbol{z} \leftharpoonup I(\boldsymbol{y}) \quad \text{such that } F(\boldsymbol{z}, \xi) = \boldsymbol{y},$$

where $\xi$ denotes a noise component. Inverse problems come in many forms and are typically broadly categorized depending on the relationship between $\boldsymbol{z}$ and $\boldsymbol{y}$. For example, in *denoising problems* $\boldsymbol{z}$ represents a "clean" version of the observed and noise-corrupted signal $\boldsymbol{y}$. In contrast, the family of problems where $\boldsymbol{z}$ represents the parameters of a system that outputs data $\boldsymbol{y}$ is typically termed *parameter estimation*.

From a statistical learning perspective, given data $D := \{\boldsymbol{y}_i\}_{i=1}^S$ and data reconstruction loss $d$, inverse problems take the following form, consisting of a reconstruction term and an optional regularization term (Adler & Öktem, 2017; Bai et al., 2020; Kamyab et al., 2022),

$$I^* := \arg\min_I \sum_{\boldsymbol{y} \in D} d(F(\boldsymbol{z}, \xi), \boldsymbol{y}) + \mathcal{R}(\boldsymbol{z}) \quad \text{where } \boldsymbol{z} \leftharpoonup I(\boldsymbol{y}). \tag{1}$$

While many inverse problem formulations assume the forward map $F$ is known, in other cases $F$ may have to be estimated alongside $I$ (and the $\arg\min$ objective optimizes both $I$ and $F$); such formulations are termed *blind* inverse problems. We additionally note that a subset of inverse problems assume $F$ to be not only known but also invertible (e.g. an invertible linear transformation) such that it renders the true latents $\boldsymbol{z}$ available in training, making learning $I$ by directly regressing on $\boldsymbol{z}$ possible. We however proceed with the standard *implicit* inverse problem learning, where the true latents $\boldsymbol{z}$ are not available, and $I$ is optimized by ensuring that the forward evaluation $F(I(\boldsymbol{y}))$ matches the observed data. Finally, inverse problems are often ill-posed, meaning that a solution may be non-existent, non-unique, or highly sensitive to the data (Adler & Öktem, 2017; Zheng et al., 2025). The regularization term $\mathcal{R}$ both incorporates any relevant priors over the latents $\boldsymbol{z} = I(\boldsymbol{y})$, and helps with ill-posedness by restricting the hypothesis space.

### 2.2 RELATED WORK

Here, we overview related work. Further related work on tasks under the GraIP framework are provided in Appendix A.1.

**Deep learning for inverse problems** Classical approaches to inverse problems require combining analytical methods with domain-specific knowledge and priors for each problem (Kamyab et al., 2022). Deep learning methods have emerged as a powerful tool for solving nonlinear inverse problems in recent years, thanks to their high representational capacity and adaptability to a wide variety of tasks. Such *neural* solvers also tend to operate under fewer assumptions on the problem setting than analytical methods, and are more adept at learning from noisy observations (Lucas et al., 2018). As a result, many neural frameworks, ranging from CNNs to diffusion models, have seen widespread use in solving inverse problems in recent years. For a comprehensive survey on neural solvers for inverse problems, we refer the reader to Bai et al. (2020); Lucas et al. (2018); Ongie et al. (2020). Finally, a recent work by Eliasof et al. (2025) focuses on adapting classical regularization techniques from inverse problems to graph settings, making it a valuable complementary contribution to ours.

**MPNNs and GTs** MPNNs have emerged as a flexible framework for machine learning on graphs and relational data, utilizing a local message-passing mechanism to learn vector representations of

graph-structured data. Notable instances of this architecture include, e.g., Gilmer et al. (2017); Hamilton et al. (2017); Velickovic et al. (2018), and the spectral approaches proposed in, e.g., Bruna et al. (2014); Defferrard et al. (2016); Kipf & Welling (2017)—all of which descend from early work in (Kireev, 1995; Scarselli et al., 2009). Besides, transformer-based models (GTs) have also attained great success on graphs, thanks to their flexibility and global information aggregation capabilities (Müller et al., 2023).

**Network inference** A complementary line of work studies network inference (also called network reconstruction), where the objective is to recover latent edge structure from indirect observations such as node signals, dynamics, or sampled interactions. Graph signal processing provides principled formulations for identifying topology from observations, accompanied by guarantees and algorithms for sparse, smooth, or diffusion-generated signals (Mateos et al., 2019). From a statistical modeling perspective, minimum description length approaches pose reconstruction as selecting the network that best compresses the data given a generative model (Peixoto, 2025b). More broadly, recent work bridges data and theory via likelihood-based inference on generative network models such as stochastic block models and variants, providing a unifying statistical framework for reconstructing networks (Peel et al., 2022). While GraIP shares with these methods the high-level goal of inferring structure from indirect data, its formulation differs in two crucial aspects. First, the inverse map in GraIP is parameterized by neural networks rather than fixed statistical estimators. Secondly, the forward map in GraIP is designed to be learnable and differentiable in most cases, which makes end-to-end gradient-based optimization feasible. Many existing formulations in network inference instead rely on discrete search or combinatorial optimization.

## 3 THE NEURAL GRAPH INVERSE PROBLEM (GRAIP) FRAMEWORK

Let us begin by considering conventional supervised graph learning tasks. We assume a (finite) set of *data* $D := \{(G_i, \boldsymbol{X}_i, y_i)\}_{i=1}^S \subseteq \mathcal{G} \times \mathbb{R}^{n \times d} \times \mathcal{Y}$, where each data point consists of a graph $G$, associated $d$-dimensional, real-valued vertex features $\boldsymbol{X}$, and target $y$. In supervised graph learning, we aim to learn some function $F : \mathcal{G} \times \mathbb{R}^{n \times d} \to \mathcal{Y}$ in order to estimate the target $y$. We term $F$ the *forward map*; $F$ is typically expected to be permutation-equivariant or -invariant (for vertex and graph-level tasks, respectively), and thus can be modeled by an MPNN or graph transformer (GT) parametrized by $\boldsymbol{\varphi}$. The objective of supervised graph learning can then be written as

$$\boldsymbol{\varphi}^* := \arg\min_{\boldsymbol{\varphi} \in \Phi} 1/|D| \sum_{(G,\boldsymbol{X},y) \in D} d\Big(F^{(\boldsymbol{\varphi})}(G, \boldsymbol{X}), y\Big),$$

Here, $d : \mathcal{Y} \times \mathcal{Y} \to \mathbb{R}^+$ is a distance measure, formally a (pseudo-)metric between elements in $\mathcal{Y}$, e.g., the 2-norm of the difference of elements in $\mathbb{R}^e$, assuming $\mathcal{Y} = \mathbb{R}^e$ for $e \in \mathbb{N}$. This setup serves as an overall blueprint of supervised graph learning, and many extensions of the proposed setup that consider edge features, edge-level tasks, and self-supervision (e.g., in the absence of labels $y$) exist.

The defining characteristic of *graph inverse problem learning* is the existence of a learnable *inverse map* $I^{(\boldsymbol{\theta})} : \mathcal{G} \times \mathbb{R}^{n \times d} \times \mathcal{Y} \to \mathcal{G}$ in addition to the forward map. In doing so, we follow the original inverse problem formulation in Equation 1, with the additional constraint that the latent $\boldsymbol{z}$ is a graph. Recall that the forward map $F$ takes in a (attributed) graph and predicts target "observations" $y$. The inverse map operates in the opposite direction to solve the *inverse problem*. That is, given target observations $y$, features $\boldsymbol{X}$, and an optional graph prior $G$, $I^{(\boldsymbol{\theta})}$ learns to "reverse-engineer" the optimal latent graph structure $\tilde{G}$ that produces this target $y$ when passed through $F$. The inverse map $I^{(\boldsymbol{\theta})}$ is typically under the same permutation-equivariance or -invariance constraints as $F$, and thus is commonly modeled using MPNNs or GTs. The formulation for GraIP then amounts to finding parameters $\boldsymbol{\theta}^* \in \Theta$ such that

$$\boldsymbol{\theta}^* := \arg\min_{\boldsymbol{\theta} \in \Theta} 1/|D| \sum_{(G,\boldsymbol{X},y) \in D} d\Big(F\Big(I^{(\boldsymbol{\theta})}(G, \boldsymbol{X}, y)\Big), y\Big) + \mathcal{R}\Big(I^{(\boldsymbol{\theta})}(G, \boldsymbol{X}, y)\Big).$$

Note that the case presented where $I^{(\boldsymbol{\theta})}$ takes all three variables $G$, $X$, and $y$ as inputs should be understood as the most general setting; in most instances of GraIP, only a subset of these inputs is used. Finally, the GraIP framing above assumes a non-blind problem with access to the forward map $F$. This is viable in specific problems such as vertex-subset problems in combinatorial optimization, where forward maps involve counting the cardinalities of sets. For blind GraIPs such as graph rewiring (see subsection 4.3), the inverse and forward maps are optimized jointly, i.e.,

$$\boldsymbol{\theta}^*, \boldsymbol{\varphi}^* := \arg\min_{\boldsymbol{\varphi} \in \Phi, \boldsymbol{\theta} \in \Theta} 1/|D| \sum_{(G,\boldsymbol{X},y) \in D} d\Big(F^{(\boldsymbol{\varphi})}\Big(I^{(\boldsymbol{\theta})}(G, \boldsymbol{X}, y)\Big), y\Big) + \mathcal{R}\Big(I^{(\boldsymbol{\theta})}(G, \boldsymbol{X}, y)\Big). \quad (2)$$

The GraIP framework is flexible enough to encompass a wide range of methods. At a high level, the requirements are minimal, (1) the inverse map $I^{(\theta)}$ must produce a graph (either by proposing one directly or by modifying an existing graph), (2) the forward map $F$ must use this graph to make predictions as in conventional (self-)supervised graph learning, and (3) the overall system must remain end-to-end differentiable for training. In what follows, we outline concrete strategies for instantiating the inverse and forward maps within the GraIP framework.

## 3.1 INVERSE MAP IN DETAIL

We outline the key aspects that guide the design of inverse maps in graph inverse problems, providing a broad characterization of the models used in our studies. The inverse map $I^{(\theta)}$ takes as input a triple $(G, \boldsymbol{X}, y)$ and outputs a graph $\tilde{G}$ as an (approximate) solution to the inverse problem. In principle, $I^{(\theta)}$ may be instantiated with any differentiable method capable of generating graph structures.

**Prior-generating models** A core component of every GraIP inverse map is a learnable, parameterized function that outputs a prior $\boldsymbol{\theta}$. This prior assigns weights to nodes or edges, where larger weights indicate a higher likelihood of inclusion in the output graph. These weighted scores are then used to construct the proposal graph $\tilde{G}$ for the forward map. More generally, the prior need not be restricted to simple edge and node scoring functions but can also serve as a parameterization of a discrete probability distribution over graphs, capturing higher-order structural dependencies beyond independent node or edge scores. To ensure permutation equivariance, the structural prior can be modeled with standard MPNNs or graph transformers, or with task-specific architectures designed to encode problem-specific inductive biases.

**Discretization and gradient-estimation strategies** In many cases, the goal is to recover a sparse graph rather than a fully connected weighted one, and some forward operators explicitly require discrete inputs. Discretization functions map the learned continuous priors to a discrete graph, typically via thresholding, non-parametric decoders, or more principled approaches that sample from a discrete exponential-family distribution parameterized by the priors and constrained by structural requirements (e.g., exactly $k$ edges or DAG constraints). These strategies, however, typically render the inverse map $I^{(\theta)}$ non-differentiable or result in zero gradients almost everywhere with respect to $\boldsymbol{\theta}$. To address this, gradient estimators such as the score-function estimator (Williams, 1992), the straight-through estimator (STE) (Bengio et al., 2013), Gumbel-softmax (Jang et al., 2016; Maddison et al., 2017), or I-MLE (Niepert et al., 2021) are commonly used, enabling differentiation through the discretization step.

Nevertheless, combining discretization with approximate gradient estimation can destabilize training and degrade outputs of inverse maps. It is therefore sometimes preferable to relax the requirement to produce a discrete graph during training. An important insight from our framework is that discretization can be harmful, by impairing stability and convergence, but also beneficial, by enforcing useful structural priors early in learning. This highlights the need for further methodological advances to better understand and control the impact of discretization in inverse graph-learning pipelines.

## 3.2 FORWARD MAP IN DETAIL

The forward map $F$ takes the graph returned by the inverse map $I^{(\theta)}$ as input to predict the target $\boldsymbol{y}$. Depending on the problem, one may have access to the true $F$ or an approximation: for example, in system dynamics simulation, one may have access to a simulator which forgoes the need to learn $F$. Many applications, however, require learning $F^{(\varphi)}$: e.g. in data-driven rewiring, the forward map is typically implemented as an MPNN predicting the downstream task on the graph rewired by the inverse map. In general, MPNNs are thus suitable forward maps for graph-based dynamics in the absence of a simulator, as message-passing over the learned interaction graph mimics the generative process.

## 4 INSTANTIATIONS OF THE GRAIP FRAMEWORK

We next present instantiations of the GraIP framework to illustrate its generality and applicability across diverse tasks; we cover two additional tasks, namely combinatorial optimization (CO) and gene regulatory network (GRN) inference, in Appendix B. For each task, we first state the problem formulation, then describe how GraIP is instantiated by specifying the role of the inverse map $I^{(\theta)}$—its inputs and intermediate outputs —the forward map $F$, and the distance measure $d$. We also denote any regularization where applicable.

Finally, for each domain, we define a baseline method based on prior work that integrates both an inverse and a forward map, as described in Section 3, and explain how we implement them. When applicable, we include a discretization strategy within the inverse map. To highlight the transferability of our framework across domains, and when applicable, we use I-MLE (Niepert et al., 2021) as the underlying discretization method in combination with an appropriate algorithm. The forward map remains domain-specific. Importantly, our goal is not to introduce new methods, but to demonstrate either *how* existing graph learning techniques can be integrated into the GraIP framework, or *how* simple baselines can be instantiated within it. Summary tables for all problems and methods considered are provided in Appendix E.

## 4.1 CAUSAL DISCOVERY (CD)

We consider the task of Bayesian network structure learning. We are given a matrix of samples $\boldsymbol{X} \in \mathbb{R}^{s \times n}$, and we assume that each of the $s$ samples is the realization of a random vector $(\boldsymbol{X}_1, \ldots, \boldsymbol{X}_n)$. Each $\boldsymbol{X}_i$ corresponds to a node in a directed acyclic graph (DAG) $G = (V, E), |V| = n$, in which edges encode dependencies. We denote $X_i^k$ the realization of $\boldsymbol{X}_i$ in the $k$-th sample. The goal is to infer the underlying DAG $G$, which is not observed during training, so we frame this task in the unsupervised setting.

**GraIP instantiation** Causal discovery (CD) naturally fits into the GraIP framework as follows: the inverse map $I^{(\boldsymbol{\theta})} \colon \mathbb{N} \to \{0,1\}^{n \times n}$ takes a vertex set $V$ with no edges, and outputs a discretized DAG $\tilde{G}$. In this setup, we assume that, for each child node $i$, $X_i^k$ has been generated by aggregating the features of the parents of $i$. We therefore assume the following parametrized forward map $F^{(\boldsymbol{\varphi})}$, for each $k \in [s]$, where $\boldsymbol{\varphi}$ denotes a set of edge weights, and $F^{(\boldsymbol{\varphi})}$ aggregates the parent's node features according to $G$ and $\boldsymbol{\varphi}$,

$$\boldsymbol{X}^k = F^{(\boldsymbol{\varphi})}(\tilde{G}, \varnothing), \qquad X_i^k = \sum_{j \in \text{parents(i)}} \varphi_j X_j^k, \tag{3}$$

where parents($i$) are the nodes of $G$ with an outgoing edge to $i$. Finally, we define the distance $d$ as the Frobenius norm between $F^{(\boldsymbol{\varphi})}$'s output and the ground-truth node features $\boldsymbol{X}$,

$$\boldsymbol{\theta}^* := \arg\min_{\boldsymbol{\theta} \in \Theta} \frac{1}{sn} \sum_{k \in [s]} \sum_{i \in [n]} d(F_t(I^{(\boldsymbol{\theta})}(\varnothing))_i, X_i^k) = \arg\min_{\boldsymbol{\theta} \in \Theta} \frac{1}{sn} \sum_{k \in [s]} \sum_{i \in [n]} \|\hat{X}_i^k - X_i^k\|_2$$

where $\hat{X}_i^k$ is the prediction of $F^{(\boldsymbol{\varphi})}$ for $X_i^k$.

BENCHMARK AND EMPIRICAL INSIGHTS

**Data** We evaluate our baseline in the setting proposed by Wren et al. (2022), generating Erdős–Rényi (ER) (Erdős & Rényi, 1960) and Barabási–Albert (BA) (Albert & Barabási, 2002) graphs and then turning them into DAGs. We generate 24 graphs for both graph types, then create node features using a Gaussian equal-variance linear additive noise model. We consider eight graph dataset configurations based on graph type $\in$ {BA, ER}, graph size $\in$ {30, 100}, and degree parameter $\in$ {2, 4}. For instance, ER2-30 denotes an ER graph with 30 nodes and an expected degree of 2, used as the ground-truth DAG. More information regarding data generation is available in Appendix C.1.

**Methods and empirical insights** We implement our main GraIP baseline, Max-DAG I-MLE, using a *discretizing* strategy, and compare it with methods based on continuous relaxations. The inverse map $I^{(\boldsymbol{\theta})}$ is a learnable prior $\boldsymbol{\theta} \in \mathbb{R}^{n \times n}$ with I-MLE as the discretization algorithm. We use a maximum DAG solver within I-MLE, namely the Greedy Feedback Arc Set (Eades et al., 1993), to ensure that the proposal graph is a DAG. The forward map is defined as a 1-layer GNN that learns a matrix of edge weights $\boldsymbol{\varphi} \in \mathbb{R}^{n \times n}$. It produces node-level predictions $X_i^k$ according to Equation 3, using the edge weights consistently with the graph produced by $I^{(\boldsymbol{\theta})}$. We evaluate this discretizing baseline against two popular, *non-discretizing* methods for DAG structure learning, NoTears (Zheng et al., 2018) and GOLEM (Ng et al., 2020), which both formulate structure learning through continuous relaxation. Since the task is to infer the ground truth DAG, we frame this as a binary classification task on the adjacency matrix. We consider several metrics, namely the F1-score, the Structural Hamming Distance (SHD), Area Under the Receiver Operating Characteristic Curve (ROC-AUC).

The results are reported in Table 1. NoTears consistently outperforms both GOLEM and the I-MLE-based method (Max-DAG I-MLE), which utilizes discretization during training. Notably, Max-DAG I-MLE performs significantly worse than these continuous approaches in most settings, underscoring the challenges of learning DAGs without constant relaxations.

Table 1: Performance comparison of NoTears, Golem, and Max-DAG I-MLE across different graph generation algorithms and densities. We report the mean over 24 ground truth graphs, as well 95% confidence intervals.

| Approach | Method | ER2-30 (top) & ER4-30 (bottom) | | | SF2-30 (top) & SF4-30 (bottom) | | |
|---|---|---|---|---|---|---|---|
| | | F1 ↑ | SHD ↓ | ROC-AUC ↑ | F1 ↑ | SHD ↓ | ROC-AUC ↑ |
| Non-discretized | NoTears | $98.5 \pm 1.8$ | $0.8 \pm 0.5$ | $99.0 \pm 0.6$ | $93.0 \pm 4.5$ | $0 \pm 0$ | $100.0 \pm 0.0$ |
| | Golem | $81.8 \pm 8.1$ | $12.5 \pm 2.8$ | $95.9 \pm 1.0$ | $91.2 \pm 4.2$ | $12.6 \pm 3.4$ | $97.8 \pm 0.9$ |
| Discretized | Max-DAG I-MLE | $94.2 \pm 1.8$ | $2.4 \pm 0.3$ | $97.1 \pm 0.4$ | $53.7 \pm 10.8$ | $22.7 \pm 2.2$ | $71.4 \pm 2.4$ |
| Non-discretized | NoTears | $100.0 \pm 0.0$ | $7.9 \pm 3.0$ | $93.4 \pm 2.2$ | $96.6 \pm 2.8$ | $3.0 \pm 1.8$ | $98.4 \pm 0.8$ |
| | Golem | $82.6 \pm 10.1$ | $10.3 \pm 2.8$ | $97.0 \pm 0.8$ | $90.8 \pm 5.9$ | $11.7 \pm 3.0$ | $98.7 \pm 0.5$ |
| Discretized | Max-DAG I-MLE | $51.9 \pm 9.8$ | $62.2 \pm 18.4$ | $73.9 \pm 2.7$ | $38.6 \pm 7.2$ | $56.7 \pm 4.2$ | $64.8 \pm 1.4$ |

## 4.2 NEURAL RELATIONAL INFERENCE (NRI)

NRI aims to infer explicit interaction structures from observations of a dynamical system, while simultaneously learning the temporal dynamics conditioned on the inferred interaction structure. For a system of $N$ objects with $d$ features over $t$ time steps

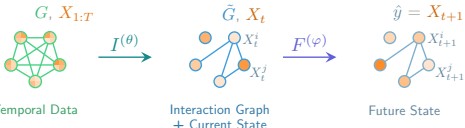

$\boldsymbol{X} = (\mathbf{x}^1, \ldots, \mathbf{x}^T) \in \mathbb{R}^{N \times d \times T}$, the goal is to find the binary or categorical relationships within the dynamical system, taking the form of an edge prediction (or classification in the categorical case) task over a graph $G$ which is optimized such that the inferred structure best explains and drives the observed system dynamics.

**GraIP instantiation** The general NRI model proposed by Kipf et al. (2018) is formulated as a variational autoencoder (VAE) that fits the GraIP framework perfectly: given temporal features $\boldsymbol{X}$ and a complete graph $G$, the inverse map $I^{(\boldsymbol{\theta})}$ consists of (1) the VAE encoder $q_\theta$ which learns a probability distribution over the edges, and (2) the sampler that obtains an interaction graph $\tilde{G}$ from the learned distribution. The forward map $F^{(\boldsymbol{\varphi})}$ implements the decoder $p_\varphi$, which uses $\tilde{G}$ to simulate the system dynamics for the next time step as a node regression task. To avoid divergence over long-horizon predictions, the forward map makes multiple forward passes to predict $M$ time steps into the future. It accumulates the errors before each gradient-based optimization step.

The pseudo-metric $d$ is primarily defined by a reconstruction error term, and a KL term for a uniform prior (following the ELBO-maximizing VAE formulation), defined as the sum of entropies, can be added as a regularizer $\mathcal{R}$ on the edge probabilities, e.g., to enforce sparsity,

$$\sum_j \sum_{t=2}^{T} \frac{\|\mathbf{x}_j^t - \hat{\mathbf{x}}_j^t\|^2}{2\sigma^2} \quad \left( -\sum_{i \neq j} H(q_\theta(\mathbf{z}_{ij}|\boldsymbol{X})) \right)$$

NRI problems come in many forms, all of which fit the GraIP framework. Kipf et al. (2018) consider both using an explicit integrator as the forward map $F$ and learning a parametrized GNN-based simulator $F^{(\boldsymbol{\varphi})}$ jointly with the inverse map. Bhaskar et al. (2024) relax the binary edge assumption to learn continuous weights over a complete graph, while Graber & Schwing (2020) relax the assumption that the interaction graph is constant across time steps to propose *dynamic* NRI to model a broader array of inverse problems. One practical extension of dNRI is gene regulatory network (GRN) inference, which we explore as a GraIP in Appendix B.2, where we aim to learn complex dynamic relationships between transcription factors, DNA, RNA, and proteins in the form of a regulatory graph.

BENCHMARK AND EMPIRICAL INSIGHTS

**Data** We focus on the Springs and Charged benchmarks (Kipf et al., 2018). In Springs, each data point is a 50-step simulation of $N \in \{5, 10\}$ objects moving in a box with random initial positions and velocities, and every pair of objects having 0.5 probability of being connected with a spring and interacting based on Hooke's law. In Charged, each object now carries a positive or negative charge, and the goal is to predict whether each node pair attracts or repels. This represents a harder task with more inherent noise. We assume a static binary interaction graph for both cases; the binary nature of the problem thus does not admit non-discretized methods. The inverse map learns the true interaction graph, while the forward map aims to accurately simulate the dynamics by message-passing over the learned graph.

**Methods and empirical insights** We use the NRI-GNN model for both the VAE encoder and decoder, which attains excellent performance on Springs, and is also highly competent on Charged. This model employs node-to-edge $(v \to e)$ and edge-to-node $(e \to v)$ message passing to learn both node- and edge-level representations effectively, and is more performant than conventional MPNN architectures for NRI tasks. We, however, note that any GNN-based model that conforms to the VAE formulation is inherently a GraIP model. The inverse map consists of the NRI encoder and the sampler. We benchmark two inverse maps: Both use the same NRI encoder and discretize via thresholding, but one uses straight-through Gumbel softmax (Jang et al., 2016; Maddison et al., 2017) (STE) for gradient estimation in the discretization step, while the other uses I-MLE. The forward map is implemented by the NRI-GNN-based decoder for the blind settings, and by the differentiable simulators provided by Kipf et al. (2018) for the non-blind ones. We report accuracy, F1-score, and ROC-AUC to evaluate the recovered graphs $\tilde{G}$. We see that on Springs both blind methods solve the task almost perfectly for $N = 5$, and are highly competent for $N = 10$. STE proves more robust for $N = 10$, though both gradient estimation methods exhibit sensitivity to thresholds as indicated by lower ROC-AUC scores.

When comparing blind and non-blind settings for the NRI + STE model, we see that the non-blind setting also solves Springs perfectly, and is particularly robust to the set threshold level for $N = 10$ (99.6% ROC-AUC) where the some sensitivity is evident in the blind case (74.9% AUC despite 98+% accuracy & F1 for a 0.5 threshold). However, on Charged, the non-blind model struggles severely with slightly above random graph metrics, whereas the the blind case is still successful (AUC: ~91% for $N = 5$, ~80% for $N = 10$). We note that the non-blind results fail here due to the vanishing gradients caused by instabilities in the simulation decoder (as per Kipf et al. (2018)), which prevents the inverse map from converging to the correct graph. This setting then showcases an "edge case" where learning in the blind setting proves more reliable than using a known forward map in the non-blind setting.

Table 2: Neural relational inference results for the **Springs** and **Charged** benchmarks, evaluating different discretization strategies on the NRI-GNN model (Kipf et al., 2018). The non-blind case refers to learning only the inverse map, while using a differentiable simulator for the forward map. We report the mean ± standard deviation reported over five seeds.

| | N | Method | Downstream Metric | Graph Metrics (%) | | |
|---|---|---|---|---|---|---|
| | | | MSE ↓ | Accuracy ↑ | F1-score ↑ | ROC-AUC ↑ |
| **Springs** | 5 | NRI + STE | 1.9e-4 ± 0.0 | 99.4 ± 0.3 | 99.3 ± 0.3 | 99.9 ± 0.0 |
| | | NRI + I-MLE | 3.2e-4 ± 0.0 | 99.5 ± 0.0 | 99.4 ± 0.1 | 100. ± 0.0 |
| | | NRI + STE (non-blind) | 1.7e-7 ± 0.0 | 99.8 ± 0.0 | 99.8 ± 0.0 | 100. ± 0.0 |
| | 10 | NRI + STE | 3.3e-5 ± 0.0 | 98.4 ± 0.0 | 98.2 ± 0.0 | 74.9 ± 0.3 |
| | | NRI + I-MLE | 1.2e-4 ± 0.0 | 91.6 ± 0.2 | 91.1 ± 0.2 | 73.4 ± 0.1 |
| | | NRI + STE (non-blind) | 1.2e-6 ± 0.0 | 98.2 ± 0.1 | 98.2 ± 0.1 | 99.6 ± 0.0 |
| **Charged** | 5 | NRI + STE | 1.2e-3 ± 0.0 | 82.8 ± 0.1 | 82.5 ± 0.1 | 91.0 ± 0.8 |
| | | NRI + STE (non-blind) | 1.8e-4 ± 0.0 | 52.6 ± 2.4 | 42.3 ± 7.0 | 57.6 ± 3.6 |
| | 10 | NRI + STE | 1.6e-3 ± 0.0 | 70.8 ± 0.7 | 70.2 ± 0.8 | 80.4 ± 1.0 |
| | | NRI + STE (non-blind) | 7.5e-4 ± 0.0 | 53.4 ± 2.1 | 49.3 ± 7.3 | 56.5 ± 2.7 |

### 4.3 DATA-DRIVEN REWIRING

*Graph rewiring* refines a given graph $G$, which may contain noise, missing or spurious edges, or structural inefficiencies. It leverages supervised learning signals as a proxy to guide the modification of $G$, producing a refined graph $\tilde{G}$ that better supports information

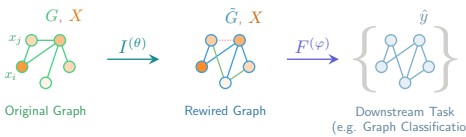

propagation and feature aggregation. This improves downstream tasks such as classification or regression, while addressing issues like over-smoothing and over-squashing through selective edge editing and optimized message passing in MPNNs. Viewed this way, graph rewiring naturally aligns with the perspective of inverse problems while bridging the gap between GraIPs and graph structure learning.

**GraIP instantiation** We build on recent approaches (Qian et al., 2023; 2024) to align graph rewiring closely with our framework: The original, noisy graph $G$, as well as the associated node features $X$, are fed to the inverse map $I^{(\theta)} \colon \mathcal{G} \times \mathbb{R}^{n \times d} \to \mathcal{G} \times \mathbb{R}^{n \times d}$, which outputs an improved graph $\tilde{G}$, while retaining the original node features $X$. The parametrized forward map $F^{(\varphi)}$ performs a downstream task,

such as graph regression or link prediction. Intuitively, instead of solely optimizing $F^{(\varphi)}$ to perform those tasks, we rewire the graph so that $F^{(\varphi)}$ can better minimize the empirical risk associated with the downstream task. In this GraIP instance, this empirical risk serves as our distance $d$. Formally, we aim to solve the following optimization problem in Equation (2). Data-driven rewiring instances typically impose a stronger structural prior on $\tilde{G}$; instead of an empty, complete or partial graph, $I^{(\theta)}$ starts with the true, known graph, and modifies it to help $F^{(\varphi)}$ learn the downstream objective better. Additionally, given the absence of a "ground truth graph" and the downstream objective being a graph learning task in itself, data-driven rewiring tasks almost always constitute blind GraIPs.

BENCHMARK AND EMPIRICAL INSIGHTS

Table 3: Comparisons between PR-MPNN with different discretization strategies. "Baseline" refers to the original GINE performance with no rewiring, whereas "Random rewire" refers to rewiring a random subset of edges within the same rewiring budget as PR-MPNN. For the WebKB datasets (Cornell/Texas/Wisconsin), mean accuracy is reported.

| Method | ZINC (MAE ↓) | Cornell ↑ | Texas ↑ | Wisconsin ↑ | Peptides-func (AP ↑) | Peptides-struct (MAE ↓) |
|---|---|---|---|---|---|---|
| Baseline (GINE) | $0.209 \pm 0.005$ | $0.574 \pm 0.006$ | $0.674 \pm 0.010$ | $0.697 \pm 0.013$ | $0.550 \pm 0.008$ | $0.355 \pm 0.005$ |
| Random rewire | $0.190 \pm 0.007$ | $0.510 \pm 0.057$ | $0.738 \pm 0.012$ | $0.731 \pm 0.005$ | $0.651 \pm 0.003$ | $0.251 \pm 0.001$ |
| Gumbel | $0.160 \pm 0.006$ | | | | | |
| I-MLE | $\mathbf{0.148} \pm 0.008$ | | | | | |
| SIMPLE | $\mathbf{0.151} \pm 0.001$ | $\mathbf{0.659} \pm 0.040$ | $\mathbf{0.827} \pm 0.032$ | $\mathbf{0.750} \pm 0.015$ | $\mathbf{0.683} \pm 0.009$ | $\mathbf{0.248} \pm 0.001$ |

**Data** Because rewiring is tuned end-to-end by the task loss, the same procedure adapts automatically to arbitrary graph types and prediction objectives, ranging from molecular property prediction to social network analysis. In this study, we demonstrate its effectiveness on molecular property prediction tasks using the ZINC dataset (Irwin & Shoichet, 2005) (the commonly used subset containing 12 000 molecules with their constrained solubility regression targets), as well as the long-range graph benchmarks Peptides-func and Peptides-struct (Dwivedi et al., 2022). We additionally evaluate our methods on the WebKB datasets Cornell, Texas and Wisconsin, which represent semi-supervised node classification tasks on heterophilic graphs (Pei et al., 2020). As there are no "ground truth graphs", we do not report any graph metrics and instead use downstream performance as a proxy.

**Methods and empirical insights** We implement graph rewiring within the GraIP framework using the PR-MPNN data-driven rewiring method (Qian et al., 2023). The inverse map is a GINE backbone (Hu et al., 2019; Xu et al., 2019) that scores candidate edges, from which a differentiable $k$-subset sampler selects a subset to add to the graph. The resulting adjacency matrix is thus better aligned with the downstream task, and the sampler serves as our *discretization strategy*.

Alongside I-MLE, we evaluate two gradient estimators for sampling, the Gumbel SoftSub-ST estimator (Jang et al., 2016; Maddison et al., 2017; Xie & Ermon, 2019) and SIMPLE (Ahmed et al., 2023). The forward map is also instantiated as a GINE backbone, which operates on the modified graph to produce task predictions. Since the sampler is differentiable, gradients flow seamlessly from the loss through the forward map into the inverse map, allowing for end-to-end optimization. This setup enables the model to leverage both the data structure and task-specific signals when learning to rewire. We compare learnable rewiring against a standard GINE baseline and random rewiring. As shown in Table 3, learnable variants outperform baselines across all tasks, with I-MLE and SIMPLE achieving the most significant gains.

## 4.4 DISCUSSION AND LESSONS LEARNED

**A single universal recipe is unlikely—but MPNN+I-MLE is a strong starting point** In a unified benchmarking framework, it is natural to seek a single, consistent inverse map. In practice, however, flexibility is essential. Some problems lack meaningful continuous relaxations (e.g., NRI, rewiring), and when they exist, they are often expensive since they induce fully connected graphs. In other settings, task-specific discretization schemes are more effective, such as gradient estimators tailored to exactly-$k$ sampling. While all GraIPs admit permutation-equivariant models, architectures, and hyperparameters typically require task-specific tuning. A pragmatic takeaway is that GraIP supports diverse design choices, with MPNN+I-MLE providing a competitive and consistent baseline. Our strategy is to adopt strong architectures from the literature (e.g., NRI-GNN for NRI, PR-MPNN for rewiring) and apply I-MLE as the discretizer. This yields a principled starting point, enabling fair comparisons between discretization strategies and continuous relaxations. Preliminary results also suggest that advanced gradient estimators are particularly beneficial for problems with complex constraints, such as CD.

**Ill-posedness becomes severe for large GraIPs** CD and GRN inference highlight cases where learnable priors and I-MLE are insufficient. Interestingly, GRN and NRI share similar formulations; yet, NRI baselines nearly recover the ground-truth graphs. A key difference is scale: GRN graphs are roughly 20 times larger but come with 150 times fewer training examples. As graph size grows, the number of pathway combinations yielding the same observation $y$ increases combinatorially, amplifying non-identifiability and demanding more data or stronger regularization. The CD benchmark exhibits a similar pattern: as graph size (from 30 to 100 nodes) and density (expected degree from 2 to 4) increases, I-MLE performance drops sharply, reinforcing the role of scale in ill-posedness.

Our preliminary observations suggest that the observed drop follows a "phase transition"—edge recovery is viable up to a certain noise threshold, where recoverability almost completely collapses to approximately random performance. Additionally, discretization over graph priors in GraIPs likely exacerbates these recoverability issues due to a fundamental bias-variance trade-off in gradient estimation: unbiased estimators (e.g., the score-function estimator) tend to have high variance, whereas attempts to lower this variance (e.g., by Gumbel-softmax or I-MLE) induces a bias (Minervini et al., 2023; Titsias & Shi, 2022). For I-MLE in particular, Minervini et al. (2023) show that the finite-difference step size $\lambda$ directly trades off gradient sparsity vs. bias: as $\lambda \to 0$, the estimated gradients become zero almost everywhere (completely uninformative), whereas a larger $\lambda$ yields denser but increasingly biased gradients. In GraIP instantiations, this may interact with highly non-convex discrete objectives, and thus discretized training is prone to getting trapped in poor local optima unless the estimator is very carefully tuned. Meanwhile, continuous relaxations, e.g. NoTears and GOLEM on CD circumvent these optimization problems by forgoing discretization altogether, accounting for their superior performance particularly on larger graphs.

These observations also allow us to draw parallels between prior work on graphical model structure learning, such as the transition in recoverability with sample size observed by Lee & Hastie (2015) and the non-identifiability phenomena reported by Bento & Montanari (2009) for Ising models. Our GraIP framework thus serves both as a tool to expose common limitations of gradient estimation across GraIPs, and also as an ideal testbed for future developments in gradient estimation for discrete learning.

**Opportunities for generative modeling and alternative approaches** Most current baselines follow the MPNN+discretizer recipe, leaving substantial room for innovation. Could autoregressive or diffusion-based graph generative models, such as DiGress (Vignac et al., 2023), serve as inverse maps? Could MPNN+reinforcement learning—as used in CO—be generalized into effective sampling strategies for other GraIPs? We argue that the solution space for GraIPs remains underexplored, and our unified framework is only a first step toward systematically addressing it. Employing graph generative models as inverse maps, however, is a non-trivial task. Unlike in imaging, where diffusion models can leverage pre-trained backbones, graph diffusion models typically must be trained from scratch for each dataset. Furthermore, guidance must handle non-differentiable rewards that arise from discretizing proposal graphs before passing them through the forward map.

## 5    CONCLUSION AND THE ROAD AHEAD FOR GRAIP

To our knowledge, this work is the first to connect inverse problems—long studied in other domains—with the emerging challenges of graph machine learning. Our key contribution is a unified framework that recasts diverse tasks as *graph inverse problems* (GraIPs)—offering a shared language, exposing links between seemingly disparate methods, and enabling transfer of ideas across subfields. To spur adoption, we release a benchmark suite spanning multiple tasks, designed as a reference point and catalyst for progress.

*Significant challenges remain.* Chief among them is the discretization bottleneck—current gradient estimators (e.g., Gumbel-Softmax, I-MLE) are often biased or unstable. Hybrid methods, reinforcement learning, and probabilistic inference could make training more robust. Stronger forward models, e.g., via graph transformers or neural–symbolic hybrids, may capture global dependencies and enforce domain constraints more effectively. Other future work may explore the generalizability of GraIP solvers, such as hybrid models that are stably pre-trained in a continuous manner and then adapted to (or fine-tuned on) discrete tasks for downstream usage, or even extensions to general-purpose, foundational inverse solvers that can be adapted to task-specific graph constraints. Ultimately, GraIP points toward general-purpose graph inverse solvers—foundation models for graph-structured data—capable of transferring across domains from combinatorial optimization to causal discovery and generative modeling. We see this as a call to action to push beyond current limitations and build the next generation of graph learning systems.

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

# A   ADDITIONAL BACKGROUND

## A.1   NOTATION

Let $\mathbb{N} := \{1, 2, \dots\}$. The set $\mathbb{R}^+$ denotes the set of non-negative real numbers. For $n \in \mathbb{N}$, let $[n] := \{1, \dots, n\} \subset \mathbb{N}$. We use $\{\!\{\dots\}\!\}$ to denote multisets, i.e., the generalization of sets allowing for multiple, finitely many instances for each of its elements. An *(undirected) graph* $G$ is a pair $(V(G), E(G))$ with *finite* sets of *vertices* $V(G)$ and *edges* $E(G) \subseteq \{\{u, v\} \subseteq V(G) \mid u \neq v\}$. For ease of notation, we denote an edge $\{u, v\}$ in $E(G)$ by $(u, v)$ or $(v, u)$. The *order* of a graph $G$ is its number $|V(G)|$ of vertices. We use standard notation throughout, e.g., we denote the *neighborhood* of a node $v$ by $N(v)$, and so on. Finally, denote $\mathcal{G}$ the set of all graphs with at most $n$ vertices.

**MPNNs**   Intuitively, MPNNs learn node features, i.e., a $d$-dimensional real-valued vector, representing each node in a graph by aggregating information from its neighboring nodes. Let $\boldsymbol{G} = (G, \boldsymbol{X})$ be an $n$-order attributed graph, where $\boldsymbol{L} \in \mathbb{R}^{n \times d}$, $d > 0$, following Gilmer et al. (2017) and Scarselli et al. (2009), in each layer, $t > 0$, we update node attributes or features,

$$\boldsymbol{h}_v^{(t)} := \mathsf{UPD}^{(t)}\Big(\boldsymbol{h}_v^{(t-1)}, \mathsf{MSG}^{(t)}\big(\{\!\{\boldsymbol{h}_u^{(t-1)} \mid u \in N(v)\}\!\}\big)\Big),$$

and $\boldsymbol{h}_v^{(0)} := \boldsymbol{X}_v$, where we assume $V(G) = [n]$. Here, the *message function* $\mathsf{MSG}^{(t)}$ is a parameterized function, e.g., a neural network, mapping the multiset of neighboring node features to a single vectorial representation. We can easily adapt a message function to incorporate edge weights or multi-dimensional features. Similarly, the *update function* $\mathsf{UPD}^{(t)}$ is a parameterized function mapping the previous node features, and the output of $\mathsf{MSG}^{(t)}$ to a single vectorial representation. To adapt the parameters of the above functions, they are optimized end-to-end, typically through a variant of stochastic gradient descent, e.g., Kingma & Ba (2015), along with the parameters of a neural network used for classification or regression.

**GTs**   To alleviate the bottleneck of MPNNs, such as their limited receptive field, Graph Transformers (GTs) have been widely adopted. A GT stacks multiple attention layers interleaved with feed-forward layers. Formally, given a graph $G$ with node attributes $\boldsymbol{X} \in \mathbb{R}^{n \times d}$, we initialize the node features as $\boldsymbol{H}^{(0)} := \boldsymbol{X}$. For each attention head at layer $t > 0$, the node representations are updated as

$$\boldsymbol{H}^{(t)} := \mathrm{softmax}\left(\frac{\boldsymbol{Q}^{(t)}\boldsymbol{K}^{(t)^T}}{\sqrt{d_k}}\right)\boldsymbol{V}^{(t)}$$

where $d_k$ denotes the feature dimension, $\boldsymbol{Q}^{(t)} := \boldsymbol{H}^{(t-1)}\boldsymbol{W}_Q^{(t)}$, $\boldsymbol{K}^{(t)} := \boldsymbol{H}^{(t-1)}\boldsymbol{W}_K^{(t)}$ and $\boldsymbol{V}^{(t)} := \boldsymbol{H}^{(t-1)}\boldsymbol{W}_V^{(t)}$ are learned linear projections of $\boldsymbol{H}^{(t-1)}$. Each attention layer typically employs multiple heads, whose outputs are concatenated as $\mathsf{MultiAttn}\big(\boldsymbol{H}^{(t-1)}\big)$. This is followed by a feed-forward layer with residual connection:

$$\boldsymbol{H}^{(t)} := \mathsf{FF}^{(t)}\Big(\mathsf{MultiAttn}\Big(\boldsymbol{H}^{(t-1)}\Big) + \boldsymbol{H}^{(t-1)}\Big).$$

To better exploit the graph structure, structural information can be incorporated either as an attention bias (Ying et al., 2021) or through structural and positional encodings (Müller et al., 2023; Rampášek et al., 2022), which are added to the node features.

**Combinatorial optimization**   Early work on combinatorial optimization on graphs (Joshi et al., 2019; Karalias & Loukas, 2020) introduced MPNN-based methods for NP-hard problems such as the traveling salesperson problem, maximum clique, and minimum vertex cover. Diffusion models (Sanokowski et al., 2024; Sun & Yang, 2023) and reinforcement learning (Khalil et al., 2017; Toenshoff et al., 2021) methods have also been proposed for solving combinatorial graph problems. We refer to Cappart et al. (2021) for a thorough survey.

**Graph rewiring and structure learning**   Graph rewiring methods (Karhadkar et al., 2022; Topping et al., 2022) address limitations such as over-smoothing and over-squashing in deep MPNNs by modifying graph connectivity to enhance information propagation. Heuristic-based approaches (Barbero et al., 2024) use curvature and spectral properties to refine edges. In contrast, data-driven techniques (Qian et al., 2023; 2024) employ probabilistic models to adjust graph structure dynamically, leveraging recent advances in differentiable sampling (Ahmed et al., 2023; Niepert et al., 2021; Qiu et al., 2022). *Graph structure learning* (GSL) (Fatemi et al., 2023) shares the goal of enhancing graph structure but differs in approach. Rewiring adjusts a given graph locally, preserving its overall structure, while GSL learns an optimized graph from raw or noisy inputs. GSL is suited to scenarios lacking reliable graphs, aiming to infer meaningful relationships. Both enhance downstream performance, but rewiring focuses on efficiency over a fixed graph, whereas GSL emphasizes structural discovery.

**Temporal and dynamic graph inference**   Graph-based learning has seen significant interest in modeling temporal and dynamic interactions, particularly in biological and social networks. Neural Relational Inference (NRI, Kipf et al. (2018)) has proven successful in learning interaction graphs for physical systems using a variational graph autoencoder. Temporal GNNs (Graber & Schwing, 2020) extend standard MPNNs by incorporating recurrent structures and attention to capture time-dependent relationships.

**Data-driven causal discovery and structure learning for graphical models**   Causal discovery aims to recover *directed acyclic graphs* (DAGs) representing underlying causal relationships in data. Traditional methods (Peters et al., 2017; Spirtes et al., 2000) rely on statistical tests and constraint-based approaches, while gradient-based techniques (Wren et al., 2022; Zheng et al., 2018) allow differentiable optimization over DAGs. Addressing this problem solely with observational data is challenging, as under the faithfulness assumption, the true DAG is identifiable only up to a Markov equivalence class. Nevertheless, identifiability can be improved through interventional data (Ke et al., 2023; Lippe et al., 2022). A related problem is network reconstruction, which infers unseen interactions between system elements based only on their behavior or dynamics (Peixoto, 2025a).

# B   EXAMPLE INSTANTIATIONS OF THE GRAIP FRAMEWORK

Here, we provide two additional example instantiations of the GraIP framework: Combinatorial optimization (B.1) and gene regulatory network (GRN) inference (B.2).

## B.1   COMBINATORIAL OPTIMIZATION

Many combinatorial optimization (CO) problems, particularly *vertex-subset problems*, align naturally with the GraIP framework. Each instance is a pair $(G, \boldsymbol{X}, S)$, where $G \in \mathcal{G}$, $\boldsymbol{X} \in \mathbb{R}^{n \times d}$, and $S \subseteq 2^{V(G)}$ denotes feasible solutions. The goal is to maximize an objective function $c_{G,\boldsymbol{X}} : 2^{V(G)} \to \mathbb{R}^+$ over $S$, i.e., find $U_G^* \in S$ such that $c_{G,\boldsymbol{X}}(U_G^*)$ is maximal. We adopt an unsupervised setting, assuming $U_G^*$ is unknown during training, to avoid the expense of label generation.

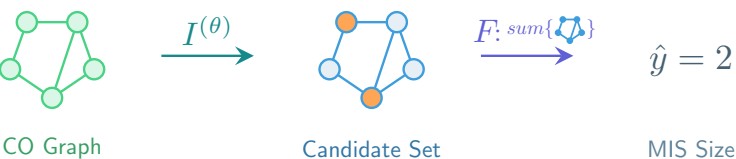

CO Graph                    Candidate Set                    MIS Size

**GraIP instantiation**   We instantiate vertex-subset problems as GraIPs as follows: The original graph $G$, as well as random walk positional encoding as its node features $\boldsymbol{X}$, are given as inputs to the inverse map $I^{(\boldsymbol{\theta})} : \mathcal{G} \times \mathbb{R}^{n \times d} \to \mathcal{G} \times \mathbb{R}^{1 \times d}$. The inverse map provides as intermediate solution the original graph structure $\tilde{G} = G$, along with soft node scores $\tilde{\boldsymbol{X}}$, which indicate membership in the solution subset. We will omit the trivial output $\tilde{G}$ for simplicity and write $\tilde{\boldsymbol{X}} = I^{(\boldsymbol{\theta})}(G, \boldsymbol{X})$.

Table 4: CO results for the maximum independent set (MIS) and maximum clique problems. Mean $\pm$ standard deviation reported for 5 seeds.

| | Non-discretized | | | | | Discretized (I-MLE) | | | |
| | MIS Size ↑ | | Max Clique Size ↑ | | | MIS Size ↑ | | Max Clique Size ↑ | |
| Method | RB-small | RB-large | RB-small | RB-large | Method | RB-small | RB-large | RB-small | RB-large |
|---|---|---|---|---|---|---|---|---|---|
| GIN | $17.65 \pm 0.0$ | $16.24 \pm 0.0$ | $14.24 \pm 0.0$ | $26.88 \pm 0.0$ | GIN | $17.40 \pm 0.6$ | $16.23 \pm 0.0$ | $14.20 \pm 0.0$ | $26.88 \pm 0.0$ |
| GCN | $17.63 \pm 0.0$ | $19.26 \pm 0.5$ | $13.82 \pm 0.4$ | $26.39 \pm 0.2$ | GCN | $\mathbf{17.67} \pm 0.1$ | $\mathbf{19.46} \pm 0.6$ | $14.01 \pm 0.1$ | $26.42 \pm 0.3$ |
| GAT | $17.46 \pm 0.2$ | $16.90 \pm 0.3$ | $13.28 \pm 0.3$ | $22.75 \pm 0.3$ | GAT | $17.43 \pm 0.1$ | $16.44 \pm 0.8$ | $12.76 \pm 0.3$ | $22.48 \pm 0.4$ |
| GCON | $16.86 \pm 0.7$ | $18.28 \pm 0.2$ | $\mathbf{15.48} \pm 0.1$ | $\mathbf{27.97} \pm 0.4$ | GCON | $17.48 \pm 0.0$ | $19.31 \pm 1.3$ | $13.79 \pm 0.2$ | $25.71 \pm 0.9$ |

Ideally, we would set the forward map to $F = c_{G,\boldsymbol{X}}$, but since $c_{G,\boldsymbol{X}}$ applies only to valid subsets $S$, and $I^{(\boldsymbol{\theta})}$ outputs soft node scores, we use a surrogate objective $\tilde{c}_{G,\boldsymbol{X}} : \mathbb{R}^{1 \times d} \to \mathbb{R}^+$, and set $F = \tilde{c}_{G,\boldsymbol{X}}$; a common strategy in MPNN approaches to CO problems, e.g., Karalias & Loukas (2020); Min et al. (2022); Wenkel et al. (2024). At test time, a non-learnable decoder $h : \mathcal{G} \times \mathbb{R}^{1 \times d} \to S$ maps node features to a valid subset $S$.

Lastly, we set $d$ to be the Frobenius norm between $\tilde{c}_{G,\boldsymbol{X}}(\tilde{\boldsymbol{X}})$ and the quality of the optimal solution w.r.t. $\tilde{c}_{G,\boldsymbol{X}}$. In the case where we want to maximize $\tilde{c}_{G,\boldsymbol{X}}$, $d\big(F(\tilde{\boldsymbol{X}}), y\big) = \big\| y - \tilde{c}_{G,\boldsymbol{X}}(\tilde{\boldsymbol{X}}) \big\|_2$ has the same minimum as the objective function $-\tilde{c}_{G,\boldsymbol{X}}\big(I^{(\boldsymbol{\theta})}(G, \boldsymbol{X})\big)$ since the output of $I^{(\boldsymbol{\theta})}$ is upper bounded by the quality of the optimal solution $y$, $y \geq \tilde{c}_{G,\boldsymbol{X}}\big(I^{(\boldsymbol{\theta})}(G, \boldsymbol{X})\big)$. The GraIP formulation is then stated as

$$\arg\min_{\boldsymbol{\theta} \in \Theta} {}^{1}\!/\!{}_{|D|} \sum_{(G,\boldsymbol{X},y) \in D} d\Big(F\Big(I^{(\boldsymbol{\theta})}(G, \boldsymbol{X})\Big), y\Big) = \arg\min_{\boldsymbol{\theta} \in \Theta} {}^{1}\!/\!{}_{|D|} \sum_{(G,\boldsymbol{X},y) \in D} -\tilde{c}_{G,\boldsymbol{X}}\Big(I^{(\boldsymbol{\theta})}(G, \boldsymbol{X})\Big).$$

For minimization problems, minimizing $d$ and $\tilde{c}_{G,\boldsymbol{X}}$ are equivalent. We discuss the surrogate-based approach in more detail, and provide two concrete example instantiations of it as GraIPs further down in Appendix B.1.

BENCHMARK AND EMPIRICAL INSIGHTS

**Data**  We focus on two CO problems in maximum independent set (MIS) and maximum clique, using synthetic RB graphs (Xu et al., 2007) derived from constraint satisfaction problem instances. We generate the graphs using the same parameters as Sanokowski et al. (2024); Wenkel et al. (2024); Zhang et al. (2023) for the datasets RB-small (200 to 300 nodes) and RB-large (800 to 1 200 nodes). For more details, please refer to Appendix C.4.

**Methods and empirical insights**  In each setting, we distinguish between *discretizing* and *non-discretizing* approaches. For non-discretizing approaches, we focus on a large family of unsupervised graph learning methods for CO, spearheaded mainly by Karalias & Loukas (2020). These unsupervised methods utilize an MPNN as an upstream model, which outputs a probability distribution over the nodes. During training, no discretization is used, meaning that the inverse map $I^{(\boldsymbol{\theta})}$ only consists of the prior model. The unsupervised surrogate objective functions that constitute the forward map $F$ are defined per CO problem and are drawn from existing literature. We refer to Appendix B.1 (further down in this section) for details on the surrogate objectives, and (Karalias & Loukas, 2020; Wenkel et al., 2024) for the test-time decoder definitions for each problem. For discretizing approaches, we round the MPNN's output to discrete assignments of membership to the solution set, and use I-MLE to differentiate through this operation.

Table 4 compares different GNN models in both setups. On MIS, discretized and non-discretized methods show similar performance, with GCN and GCON (Wenkel et al., 2024) performing best on RB-large. On the max-clique problem, GCON without discretization outperforms other methods on both graph sizes.

DETAILS AND CONCRETE INSTANTIATIONS

**Overview: Surrogate objective approach**  In MPNN-based approaches to CO problems, the *true* cost function typically applies only to valid subsets $S$. Additionally, most MPNN-based methods for vertex-subset problems (Karalias & Loukas, 2020; Min et al., 2022; Wenkel et al., 2024) operate on a

continuous relaxation of the problem for training stability, such that $I^{(\boldsymbol{\theta})}$ outputs soft node scores or *probabilities* of nodes being in the target set, rather than binary outputs.

Such formulations use a surrogate objective $\tilde{c}_{G,\boldsymbol{X}} : \mathbb{R}^{1 \times d} \to \mathbb{R}^+$, and set $F = \tilde{c}_{G,\boldsymbol{X}}$, e.g.,

$$\tilde{c}_{G,\boldsymbol{X}}(\tilde{\boldsymbol{X}}) := b_{G,\boldsymbol{X}}(\tilde{\boldsymbol{X}}) + \alpha \, q_{G,\boldsymbol{X}}(\tilde{\boldsymbol{X}}),$$

where $\alpha$ is a hyperparameter, $b_{G,\boldsymbol{X}}$ indicates the scores' fitness w.r.t. the objective function $c_{G,\boldsymbol{X}}$, and $q_{G,\boldsymbol{X}}$ softly enforces constraints, e.g., using a standard log-barrier approach. While $q_{G,\boldsymbol{X}}$ guides $I^{(\boldsymbol{\theta})}$ toward feasible solutions, it does not guarantee constraint satisfaction.

At test time, a non-learnable decoder $h \colon \mathcal{G} \times \mathbb{R}^{1 \times d} \to S$ maps node features to a valid subset $S$. We now introduce the seminal work by (Karalias & Loukas, 2020) to better understand the problem framing, and discuss two surrogate objectives for the maximum independent set (MIS) and maximum clique problems.

**Erdős goes neural**   The method proposed by Karalias & Loukas (2020) is a special case of our framework. The scores attached to each node in $M$ are interpreted as individual probabilities $p_M(v)$ that node $v$ is in the subset. This can then be used to define a probability distribution over subsets $U$ of nodes by assuming that each node's membership is drawn independently. We denote this probability distribution with $p_M(U)$. We then set

- $I^{(\boldsymbol{\theta})}$ to be an MPNN that outputs the probability for each node,
- $b_G(M) = \mathbb{E}_{U \sim p_M(U)}\big[c_G(U)\big]$, $q_G(M) = p_M(U \notin S)$, and[1]
- $h$ to be a sequential decoder as follows. First, order the nodes of $M$ in decreasing order of probability, $v_1, \ldots, v_n$. Then, let $U_s = \emptyset$ be the set of nodes that have been accepted into the solution, and let $U_r = \emptyset$ be the nodes that have been rejected. During each iteration $i$, node $v_i$ is included in $U_s$ if

$$\mathbb{E}_{U \sim p_M(U)}\Big[c_G(U) + \alpha \mathbb{1}(U \notin S) \,\Big|\, U_s \subset U, \, U \cap U_r = \emptyset, \, v_i \in U\Big]$$
$$> \mathbb{E}_{U \sim p_M(U)}\Big[c_G(U) + \alpha \mathbb{1}(U \notin S) \,\Big|\, U_s \subset U, \, U \cap U_r = \emptyset, \, v_i \notin U\Big],$$

and included in $U_r$ otherwise.[2]

In practice, surrogate objective functions specific to a particular CO problem are often easier to compute than the general choice for $b_G$ and $q_G$ described above. We will now describe such surrogate objectives for the maximum independent set problem (MIS) and the maximum clique problem.

**Surrogate objective for the maximum independent set problem**   Given an undirected graph $G$ with nodes $V(G)$ and edges $E(G)$, an independent set is defined as a subset of nodes $U \subseteq V(G)$, such that no edge connects each pair of nodes in $U$. The MIS asks for the largest independent set in a given graph. Mathematically, we aim to optimize

$$\max_{U \subseteq V(G)} |U| \qquad \text{s.t. } \forall u, v \in U : (u, v) \notin E(G).$$

Following Toenshoff et al. (2021), we can choose $q_G(M)$ based on the probability that for a given edge, at most one of the two nodes is in $U$. Here, $q_G(M)$ maximizes the combined log-likelihood over all edges,

$$q_G(M) := \frac{1}{|E(G)|} \sum_{(u,v) \in E(G)} \log\Big(1 - p_{U \sim p_M(U)}\big[u \in U, v \in U\big]\Big).$$

$b_G(M)$ is simply defined as

$$b_G(M) := \frac{1}{n} \sum_{v \in V(G)} p_M(v)$$

to maximize the number of nodes in the set. The maximization objective used for training is then

$$\tilde{c}_G(M) = \frac{1}{n} \sum_{v \in V(G)} p_M(v) + \alpha \frac{1}{|E(G)|} \sum_{(u,v) \in E(G)} \log\big(1 - p_{U \sim p(U)}\big[u \in U, v \in U\big]\big).$$

---

[1] The expectation can be replaced with a suitable upper bound if it cannot be computed in closed form.

[2] We assume here without loss of generality that the objective is a maximization objective. For minimization objectives, invert the inequality.

**Surrogate objective for the maximum clique problem**  Given an undirected graph $G$ with nodes $V(G)$ and edges $E(G)$, a clique is a subset of nodes $U \subseteq V(G)$, such that each pair of distinct nodes in $U$ is connected by an edge. The maximum clique problem asks for the largest clique in a given graph. Mathematically, this means optimizing

$$\max_{U \subseteq V(G)} |U| \qquad \text{s.t. } \forall u, v \in U, u \neq v : (u, v) \in E(G).$$

Min et al. (2022) note that finding the maximum clique is equivalent to finding the the clique with the most edges, and use this to derive the following surrogate loss for the maximum clique problem. For ease of notation, we assume $V(G) = [n]$ and write the probabilities of $p_M$ as a vector $\boldsymbol{p}$ such that $\boldsymbol{p}_v = p_M(v)$. For a given subset of nodes $U \subseteq V(G)$, the number of edges between nodes in $U$ is $\sum_{u,v \in U} \boldsymbol{A}(G)_{uv}$, where $\boldsymbol{A}(G)$ is the adjacency matrix of $G$. This can be used to define

$$b_G(M) := \mathbb{E}_{U \sim p_M(U)} \left[ \sum_{u,v \in U} \boldsymbol{A}(G)_{uv} \right] = \sum_{(u,v) \in E(G)} \boldsymbol{p}_u \boldsymbol{p}_v = \boldsymbol{p}^T \boldsymbol{A}(G) \, \boldsymbol{p}.$$

To softly enforce the constraints, Min et al. use

$$q_G(M) := \boldsymbol{p}^T \overline{\boldsymbol{A}}(G) \, \boldsymbol{p},$$

where $\overline{\boldsymbol{A}}(G) = \mathbf{1}_{n \times n} - (\boldsymbol{I} + \boldsymbol{A}(G))$ is the adjacency matrix of the complement graph. Combining these two, the surrogate maximization objective becomes

$$\tilde{c}_G(M) = \boldsymbol{p}^T \boldsymbol{A}(G) \, \boldsymbol{p} - \alpha \boldsymbol{p}^T \overline{\boldsymbol{A}}(G) \, \boldsymbol{p}.$$

## B.2 GENE REGULATORY NETWORK INFERENCE

In this section, we describe an additional instantiation of the GraIP framework, the inference of Gene Regulatory Networks (GRN). GRNs involve complex interactions among transcription factors, DNA, RNA, and proteins, which dynamically regulate each other through feedback loops, forming a dynamical system. This system is best represented by a *regulatory interaction graph*, making its inference inherently a temporal problem closely related to dynamic NRI.

Hence, we define our interaction graph as a (directed) time-varying graph $G_t \in \mathcal{G}_1$ on a fixed set of vertices $V$ with features $\boldsymbol{X}_t$. Unlike NRI, in GRN inference, we do not assume binary or categorical edges; instead, we model $\tilde{G}$ as complete with continuous edge weights. We assume that the gene expressions $\boldsymbol{X}_{t+1}$ are a function of previous expressions $\boldsymbol{X}_t$ and the weighted adjacency matrix $\boldsymbol{W}_t$ over the complete graph at time $t$, where $F_t$ is the forward map at time $t$, $F_t : \mathcal{G}_1 \times \mathbb{R}^n \to \mathbb{R}^n$:

$$\boldsymbol{X}_{t+1} = F_t(\tilde{G}_t, \boldsymbol{X}_t) = F_t(\boldsymbol{W}_t, \boldsymbol{X}_t),$$

GRN inference aims to reconstruct the regulatory interaction graph as a weighted adjacency matrix $\boldsymbol{W}_t$ given the node features $\boldsymbol{X}_{t+1}$ at each time step, and learn the gene expression dynamics for the given GRN over time. The temporal graph is then used by either a heuristic or a learned, e.g., message-passing based, simulator to run the dynamical system accurately.

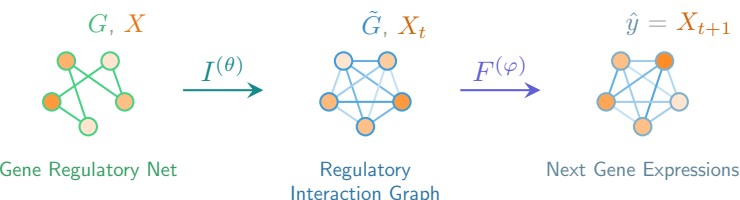

**GraIP instantiation**  The inverse map $I^{(\boldsymbol{\theta})}$ learns to reconstruct the adjacency matrix $\boldsymbol{W}_t$ given the node features $\boldsymbol{X}_{t+1}$ at time $t = 1$, and an optional prior graph $G_p$ typically derived from domain knowledge about the GRN in question. The prior graph helps with identifiability for large GRNs; in cases where a prior is absent, we set $G_p = (V, \varnothing)$.

$I^{(\boldsymbol{\theta})}$ outputs a graph with edge weights $\hat{\boldsymbol{W}}_t$ and $(\hat{\boldsymbol{W}}_t, \boldsymbol{X}_t) = I^{(\boldsymbol{\theta})}(G_p, \boldsymbol{X}_{t+1})$, which is passed further through the forward map such that $F_t(\hat{\boldsymbol{W}}_t, \boldsymbol{X}_t) \approx \boldsymbol{X}_{t+1}$. The forward map then predicts the gene expression levels for the next time step as an NRI-like node regression task.

We can define our distance $d$ as the Frobenius norm between $\hat{\boldsymbol{X}}_t$ and $\boldsymbol{X}_t$. The model predicts the GRN dynamics one time-step at a time through the following formulation, where for a time window $\mathcal{T}$ we have,

$$\boldsymbol{\theta}^* = \arg\min_{\boldsymbol{\theta} \in \Theta} 1/|D| \sum_{t \in \mathcal{T}} \sum_{(\boldsymbol{X}_t, \boldsymbol{X}_{t+1}) \in D} d(F_t(I^{(\boldsymbol{\theta})}(G_p, \boldsymbol{X}_{t+1}), \boldsymbol{X}_t), \boldsymbol{X}_{t+1})$$

$$= \arg\min_{\boldsymbol{\theta} \in \Theta} 1/|D| \sum_{t \in \mathcal{T}} \sum_{(\boldsymbol{X}_t, \boldsymbol{X}_{t+1}) \in D} d(\hat{\boldsymbol{X}}_{t+1}, \boldsymbol{X}_{t+1}).$$

Similar to NRI, the model predicts $M$ time steps into the future and accumulates the errors before each gradient-based optimization step to avoid divergence over long-horizon predictions.

BENCHMARK AND EMPIRICAL INSIGHTS

**Data**    We pose the GRN inference problem as a node regression task. That is, we consider a complete graph with $n$ nodes, where each node represents a gene, and a single node feature represents the expression level of that gene. We follow RiTINI (Bhaskar et al., 2024) in using the SERGIO simulator (Dibaeinia & Sinha, 2020) to generate temporal gene expression data derived from a GRN based on 100 genes and 300 single-cell samples. We then fit a MIOFlow model (Huguet et al., 2022) over the samples to obtain continous trajecories. We finally select and sample from five frajectories to construct our training data. We provide more details on the data generation process in Appendix C.5.

We propose two related node-regression tasks for our GraIP models for GRN inference. In the former, we train our models with the Markov assumption (to circumvent framing it as a temporal problem) such that given a data point with expression levels for 100 genes, the model aims to predict the expression levels for the next time step. The latter is a more difficult temporal learning task, where the model seeks to predict expression levels for the next five time steps. Bhaskar et al. (2024) demonstrate that utilizing a neural ODE (Chen et al., 2018) significantly improves performance for these tasks; therefore, we employ a neural ODE framework with our GraIP models in this context.

**Methods and empirical insights**    We again draw our primary method from RiTINI Bhaskar et al. (2024), which leverages an attention mechanism for GRN inference. RiTINI utilizes a single GAT layer over a graph prior in the form of a subsampled version of the ground-truth graph, aiming to recover the full graph, which makes for a more tractable problem. We slightly alter this setup and forgo the assumption of a graph prior, instead starting from a complete graph and attempting to directly infer structure from attention scores using a GAT or GT.

The use of a single attention-based layer directly fits the GraIP framework. The attention computation associated with a GAT or graph transformer layer implements the inverse map, as the attention coefficients effectively induce a prior over the complete graph. The forward map is then implemented by the "message propagation" step within the layer that follows the attention computation. This forward map uses the attention coefficients from the inverse map to perform weighted message passing effectively.

GAT consistently attains lower MSE and variance than GT, meaning it can predict the gene expression levels in further time steps more accurately (and better recover the ground truth graph), justifying its use in RiTINI. We note that the relatively small dataset with only several hundred single-cell data points likely favors GAT; we aim to provide more benchmarking studies on larger GRN datasets better to understand model (and attention prior) behavior.

**Discussion**    The framework presented here involves some simplifications over the one presented in RiTINI (Bhaskar et al., 2024) to emphasize components most pertinent to the inverse problem; these simplifications can be rectified later.

- The most fundamental difference is that we assume regularly spaced, discrete time steps $t = [0, T]$. In contrast, RiTINI is designed to handle irregular time steps, which is more suited for the continuous domain. We also note that other works, such as Dynamic Neural Relational

Table 5: GRN results for batch trajectory experiments ($k = 5$). All results are $\times 10^{-5}$. Mean $\pm$ std reported for three seeds.

| Time window (2 steps) | | Time window (5 steps) | |
|---|---|---|---|
| **Method** | **MSE** $\downarrow$ | **Method** | **MSE** $\downarrow$ |
| GAT | **75.12** $\pm$ 4.9 | Neural ODE (GAT) | **87.08** $\pm$ 27 |
| GT | 80.69 $\pm$ 5.3 | Neural ODE (GT) | 143.60 $\pm$ 45 |

Inference (dNRI) (Graber & Schwing, 2020), tackle similar problems in the discrete domain. We primarily deal with irregularly sampled data points because experimental data, such as single-cell data, often exhibit these irregularities, where different nodes are sampled at varying time points. RiTINI thus updates node features using the parents' node features from a recent $[t, t - \tau]$ window to handle this.

- Another consequence of this irregular sampling is that it makes modeling the features for the next time-step directly difficult during training, as the time difference to model forward may vary significantly. Bhaskar et al. (2024) thus uses a Neural ODE (Chen et al., 2018) with an ODE solver to extrapolate to arbitrary time instead of modeling the future state directly. Our previous assumption of uniform, discrete steps thus makes the use of a Neural ODE redundant. However, viewing the Neural ODE framework as a variant of the GraIP problem is also a viable option.

- We also assume a Markovian process, i.e., the next state $\boldsymbol{X}_{t+1}$ depends only on the current state $\boldsymbol{X}_t$. However, GRNs typically exhibit various hysteresis or lag effects that may persist across multiple time steps, meaning we may depend on some arbitrary length $\delta \in [t, t - \tau]$ into the past.

- Finally, we assume no (time-independent) graph prior for simplicity. It is, however, straightforward to incorporate a graph prior $\mathcal{P}$ as an auxiliary variable, and adding a regularization term $\mathcal{R} := \left\| \hat{\boldsymbol{W}}_t - \boldsymbol{W}_{\mathcal{P}} \right\|_{\mathrm{F}}$ to our pseudo-metric $d$ that punishes deviations from the prior,

$$F_{n,t}(\boldsymbol{X}_t, \boldsymbol{W}_t) = \boldsymbol{X}_{t+1}$$
$$F_{n,t}^{-1}(\boldsymbol{X}_{t+1}, \boldsymbol{X}_t, \mathcal{P}) = \boldsymbol{W}_t$$
$$d(\hat{\boldsymbol{X}}_t, \hat{\boldsymbol{W}}_t, \boldsymbol{X}_t, \boldsymbol{W}_{\mathcal{P}}) = \left\| \hat{\boldsymbol{X}}_t - \boldsymbol{X}_t \right\|_{\mathrm{F}} + \alpha \left\| \hat{\boldsymbol{W}}_t - \boldsymbol{W}_{\mathcal{P}} \right\|_{\mathrm{F}}.$$

# C  DATA & HYPERPARAMETERS FOR BASELINES

## C.1  CAUSAL DISCOVERY

We evaluate our baseline in the setting proposed by Wren et al. (2022). It consists of generating Erdős–Rényi (ER) and Barabási–Albert (BA) graphs and then turning them into DAGs. We generate 24 graphs for both graph types, then create node features using a Gaussian equal-variance linear additive noise model. For each random graph, we sample 1400 data points and use an 80/10/20 split.

We consider eight configurations for generating the ground-truth graph:

- Graph distribution: Erdős–Rényi (ER) or Barabási–Albert (BA);
- Graph size: 30 or 100 nodes;
- Degree parameter: 2 or 4. For ER graphs, this corresponds to the expected degree of each node; for BA graphs, it specifies the number of edges attached to each newly added node.

These three parameters identify each configuration. For instance, `ER2-30` denotes an ER graph with 30 nodes and an expected degree of 2, used as the ground-truth DAG.

## C.2  NEURAL RELATIONAL INFERENCE

We use the Springs dataset from Kipf et al. (2018) as our benchmark: $N \in \{5, 10\}$ particles are simulated in a 2D box according to Newton's laws of motion, where a given pair of particles is connected with a

spring with probability 0.5. The connected pairs exert forces on each other based on Hooke's law, such that the force applied to $v_j$ by $v_i$ is calculated as $F_{ij} = -k(r_i - r_j)$ based on particle locations $r_i, r_j$ and a given spring constant $k$. Initial location and velocities are sampled randomly. For training data, PDE-based numerical integration is applied to solve the equations of motion over 5 000 time-steps, and every 100th step is subsampled to generate training samples of 50 steps each. The inverse map attempts to learn the interacting pairs, while the forward map learns to predict location and velocity information over 50 time steps, thereby simulating Newtonian dynamics accurately. We note that while we benchmark in the *blind* inverse problem setting, it is viable to use this numerical integrator as a ground truth simulator $F$ to benchmark in the non-blind setting.

Our baseline model for both the inverse map prior and forward map is the NRI-GNN from Kipf et al. (2018). NRI-GNN is an MPNN architecture that uses both node-to-edge ($v \rightarrow e$) and edge-to-node ($e \rightarrow v$) message-passing with MLP components to learn both node and edge-level representations. It is particularly useful for NRI tasks compared to conventional MPNNs, since the edge-level representations are required for edge scoring in the inverse map, while the node-level representations are required for the downstream dynamics prediction. This allows us to use similar architectures for both maps: In our benchmarks, both the inverse map encoder and the forward map decoder consist of two message-passing steps where the edge or node representations from the last message-passing step are passed through an MLP to make the respective predictions. We fix the hidden dimension and batch size to 256, and use an AdamW optimizer (Loshchilov & Hutter, 2017) with learning rate 0.0001 for 250 epochs. We use a learning rate scheduler with 0.5 factor and 100 patience. Keeping all else the same, we benchmark two gradient estimators in STE and I-MLE.

### C.3 DATA-DRIVEN REWIRING

Our approach utilizes a two-stage architecture with an upstream and a downstream component. For the upstream model, we identify 200 potential edge candidates to be added, from which 10 are sampled. It is implemented as a GIN model with four layers and a hidden dimension of 128, outputting scores for the edge candidates. The scores are inputs to the discretizers (I-MLE, SIMPLE, or Gumbel softmax). The downstream model processes the rewired graph using a separate GIN model, also with four layers but a larger hidden dimension of 256.

We train the model to minimize the Mean Absolute Error (MAE) loss and report the final MAE on the test set. We use the Adam optimizer Kingma & Ba (2015) with a learning rate of 0.001 and no weight decay. The model is trained for a maximum of 1000 epochs. We employ early stopping with a patience of 200 epochs and use a learning rate scheduler that reduces the learning rate on a plateau with a patience of 100 epochs.

### C.4 COMBINATORIAL OPTIMIZATION

We evaluate our approach on Combinatorial Optimization (CO) problems using two datasets, small and large, each containing 40 000 RB graphs (Xu et al., 2007) generated following the procedure in Sanokowski et al. (2024); Wenkel et al. (2024); Zhang et al. (2023).

- Small graphs: Generated with a clique count in the range [20,25] and a clique size in the range [5,12].
- Large graphs: Generated with a clique count in the range [40,55] and a clique size in the range [20,25].

For both datasets, the tightness parameter is sampled from [0.3,1]. After generation, we filter the datasets to include only graphs with a specific node count: [200,300] for the small dataset and [800,1200] for the large dataset.

We evaluate four GNN architectures: GIN, GCN, GAT, and GCON. All models share a common structure of 20 GNN layers with a hidden dimension of 32. Node input features are 20-dimensional random walk positional encodings.

Models are trained using a surrogate loss as defined in Wenkel et al. (2024). We use the AdamW optimizer Loshchilov & Hutter (2017) with a learning rate of 0.001 and weight decay of 0.02. Training runs for a maximum of 100 epochs, with early stopping triggered after 20 epochs of no improvement (patience 20). We also employ a learning rate scheduler, reducing the learning rate on a plateau with a patience of 10

epochs. During inference, we decode solutions by greedily sorting the model's output scores to form either a maximal independent set or a maximum clique.

### C.5 GENE REGULATORY NETWORK INFERENCE

We follow the RiTINI (Bhaskar et al., 2024) paper in using SERGIO (Dibaeinia & Sinha, 2020) to simulate a GRN using identical parameters; the resulting dynamic GRN data represent expression levels for $n = 100$ genes across 300 single-cell samples, simulated based on a differentiation system with two branches. We then fit a MIOFlow model (Huguet et al., 2022) over these single-cell samples to obtain continuous, (pseudo)-temporal gene trajectories we can sample from. Finally, we uniformly discretize the time dimension into 38 bins to simplify the sampling process. In GRN data, branching of the samples is a common phenomenon (e.g., as cells differentiate into distinct groups over time). In the results presented, we focus on only MIOFlow-derived trajectories belonging to a single branch, as different branches induce different underlying gene interaction graphs. Five trajectories are selected and sampled to constitute the node features at each time step.

## D   REGULARIZATION FOR (GRAPH) INVERSE PROBLEMS

Avoiding degenerate solutions is one of the key considerations in solving GraIPs, and inverse problems in general. Here, we discuss this problem in the context of ill-posedness and regularization. Ill-posedness is a more general term that encompasses inverse problems with no solution or multiple solutions; degenerate solutions arise in this context as a trivial solution among multiple ones. An extensive survey on ill-posedness in inverse problems is provided by Kabanikhin (2008).

Avoiding degeneracy and ill-posedness in inverse problems is typically achieved via applying a form of regularization over the solution space. This makes regularization a key component of handling inverse problems in general, and a potential avenue for future work on GraIPs through the development of novel and/or task-specific graph regularizers. We thus aim to provide here a very brief overview of regularization techniques for classical inverse problems, their extensions to graphs in related work, and how these differ from graph regularization techniques employed in GraIPs.

In general inverse problems, there is a vast established literature on regularization, such as Tikhonov regularization (somewhat analogous to weight decay strategies in deep learning), smoothness-based regularizers, and iterative methods. We refer to Benning & Burger (2018) and Aster et al. (2019) for a comprehensive overview. Such inverse problem regularization techniques, however, are distinct from those for GraIPs, as the solutions to general inverse problems are not structured and can simply be represented in vector or matrix form.

There also exist some, albeit limited, prior works that consider regularization for inverse problems over graphs. Ling et al. (2024) uses graph diffusion priors to solve source estimation problems; Eliasof et al. (2025) then focus on a more general extension of learnable regularizers for inverse problems on graph data. In the latter work, extensions of classical regularizers to the graph domain are also discussed. To enforce smoothness over node features of the graph, for example, the graph Laplacian can be used:

$$\mathbf{R}(\mathbf{x}) = \frac{1}{2}\mathbf{x}^\top \mathbf{L}\mathbf{x}$$

If $\mathbf{L}$ is replaced by the identity $\mathbf{I}$, this becomes equivalent to Tikhonov regularization, where we simply aim to reduce the norm of the features. Note that the kind of inverse problems that Ling et al. (2024) and Eliasof et al. (2025) tackle is fundamentally different from GraIPs, despite also being defined over graphs. In their setting, one is interested in predicting node/edge features or properties from known graph structures (source estimation, graph transport, etc.). In that sense, solutions to both their forward and inverse problems are defined over node or edge features rather than graph structures. To follow up on the source estimation example, the forward map in this case is a $k$-step diffusion process defined by the transition matrix $\mathbf{P}^k$, and the goal is to identify the source node from the final node features after diffusion. The inverted process *relies* on the known graph structure, but *does not learn or optimize* it, indicating an inverse problem paradigm more akin to general IPs than the one we are interested in.

The regularization strategies we may consider in GraIPs are, on the other hand, inherently structural, and thus involve regularizing the graph itself by choosing an appropriate (e.g., sparse) prior distribution for the

graph, or introducing a loss term that penalizes the number of edges in the adjacency matrix, for example. In one can also leverage domain-specific information: In RiTINI (Bhaskar et al., 2024), for example, the authors enforce meaningful graphs for gene regulatory network (GRN) inference by using a partial graph derived from domain knowledge of the GRN in question, the task of the inverse problem then becomes "completing" this partial graph rather than proposing one from scratch, alleviating the graph identifiability problem significantly.

# E  SUMMARY TABLES FOR PROBLEMS AND BASELINE METHODS

Table 6 summarizes the problems covered by the example instantiations of GraIP that we present in this work. Table 7 lists the baselines that we evaluate on these problems.

Table 6: Summary of the problems for which we instantiate GraIP in this work.

| Problem | Discrete? | Inputs | Intermediate Solution | Inputs to $F$ | Outputs $\hat{y}$ | Distance measure $d(y, \hat{y})$ |
|---|---|---|---|---|---|---|
| **CD** | Yes | $G = (V, \varnothing)$, $\boldsymbol{X} = \varnothing$ | $\tilde{G}$: DAG | $\tilde{G}$, parent node features $y$ | All node features | Frobenius norm |
| **NRI** | Yes$^*$ | $G = (V, E_{\text{comp}})$, $\boldsymbol{X}_{1:T}$ | $\tilde{G}$: Sparsified | $\tilde{G}$, time step(s) $\boldsymbol{X}_t$ | Next time step(s) $\boldsymbol{X}_{t+1}$ | Reconstruction error (+ KL regularization) |
| **Rewiring** | Yes | $G = (V, E)$, $\boldsymbol{X}$ | $\tilde{G}$: Rewired | $\tilde{G}$, $\boldsymbol{X}$, (opt.) $G$ | Downstream target | Downstream empirical risk function |
| **CO (vertex)** | Yes$^*$ | $G = (V, E)$, $\boldsymbol{X} = \varnothing$ | $G$, prior $\tilde{\boldsymbol{X}}$ | $G, \tilde{\boldsymbol{X}}$ | Surrogate objective value | Frobenius norm (implicit) |
| **GRN** | No | $G = (V, E_{\text{comp}})$ (opt.) prior $E$, $\boldsymbol{X}_{1:T}$ | $\tilde{G}$: Reweighted | $\tilde{G}$, time step(s) $\boldsymbol{X}_t$ | Next time step(s) $\boldsymbol{X}_{t+1}$ | Frobenius norm (+ graph prior regularization) |

$^*$ with continuous relaxations possible

Table 7: Summary our baseline methods.

| Problem | Method Name | Discrete? | Prior model | Inverse map | | Forward map |
| | | | | Discretizer | | |
| | | | | Sampling | Gradient estimation | |
|---|---|---|---|---|---|---|
| CD | NoTears | No | | N/A | N/A | 1-layer GIN |
| CD | Golem | No | | N/A | N/A | 1-layer GIN |
| CD | Max-DAG-I-MLE | Yes | Learnable prior | Max-DAG | I-MLE | 1-layer GIN |
| NRI | NRI + STE | Yes | NRI-GNN encoder | Thresholding | STE | NRI-GNN decoder |
| NRI | NRI + STE (non-blind) | Yes | NRI-GNN encoder | Thresholding | STE | Differentiable simulator |
| NRI | NRI + I-MLE | Yes | NRI-GNN encoder | Thresholding | I-MLE | NRI-GNN decoder |
| Rewiring | Base | Yes | GINE | $k$-subset sampler | N/A | GINE |
| Rewiring | Rand Rewire | Yes | GINE | $k$-subset sampler | N/A | GINE |
| Rewiring | Gumbel | Yes | GINE | $k$-subset sampler | Gumbel | GINE |
| Rewiring | I-MLE | Yes | GINE | $k$-subset sampler | I-MLE | GINE |
| Rewiring | SIMPLE | Yes | GINE | $k$-subset sampler | SIMPLE | GINE |
| CO | GIN (Non-disc.) | No | GIN | N/A | N/A | Surrogate CO objective |
| CO | GCN (Non-disc.) | No | GCN | N/A | N/A | Surrogate CO objective |
| CO | GAT (Non-disc.) | No | GAT | N/A | N/A | Surrogate CO objective |
| CO | GCON (Non-disc.) | No | GCON | N/A | N/A | Surrogate CO objective |
| CO | GIN (Disc.) | Yes | GIN | Thresholding | I-MLE | Surrogate CO objective |
| CO | GCN (Disc.) | Yes | GCN | Thresholding | I-MLE | Surrogate CO objective |
| CO | GAT (Disc.) | Yes | GAT | Thresholding | I-MLE | Surrogate CO objective |
| CO | GCON (Disc.) | Yes | GCON | Thresholding | I-MLE | Surrogate CO objective |
| GRN | GAT | No | 1-layer GAT | N/A | N/A | 1-layer GAT |
| GRN | GT | No | 1-layer GT | N/A | N/A | 1-layer GT |
| GRN | Neural ODE (GAT) | No | 1-layer GAT + Neural ODE | N/A | N/A | 1-layer GAT + Neural ODE |
| GRN | Neural ODE (GT) | No | 1-layer GT + Neural ODE | N/A | N/A | 1-layer GT + Neural ODE |

