# OpenReview forum: "GraIP: A Benchmarking Framework for Neural Graph Inverse Problems"
_ICLR.cc/2026/Conference — Submitted to ICLR 2026_

### Official Review · Reviewer_JtZo · 2025-10-26

**Soundness:** 2
**Presentation:** 3
**Contribution:** 2
**Rating:** 4
**Confidence:** 4

**Summary:**

The paper proposes a unified and general framework for casting graph learning problems as *inverse problems on graphs*. The main idea is that many graph learning tasks can be viewed as finding an inverse mapping that explains a forward process (e.g., diffusion dynamics, biological interactions), thereby recovering the underlying graph structure that generated the observations. The proposed *Neural Graph Inverse Problem (GraIP)* framework formalizes this idea and is instantiated across several applications, including causal discovery, neural relational inference, and graph rewiring. For each task, the authors benchmark existing methods and discretization strategies, demonstrating both the generality and challenges of the framework.

Overall, this is a well-written and timely paper that makes a valuable conceptual and benchmarking contribution. Unifying graph-learning tasks of different natures under the inverse-problem paradigm is novel and potentially impactful. However, there remains a noticeable misalignment between the theoretical formulation and its practical instantiations, and the experimental analysis is somewhat shallow in explaining and interpreting results. Addressing these issues would significantly strengthen the paper.

**Strengths:**

- **Timeliness and relevance:** The paper tackles an important and underexplored topic, aiming to fill a gap in the graph learning literature by bringing insights from inverse problems in imaging and physics-based domains to graph-structured data.
- **Generality and scope:** The framework is applied to several distinct graph learning problems (causal discovery, neural relational inference, and graph rewiring), supporting the claim of generality.
- **Clarity and writing quality:** The paper is clearly written and well structured, making it easy to follow.
- **Benchmarking contribution:** The inclusion of multiple tasks and comparison of discretization strategies (I-MLE, Gumbel, SIMPLE) is challenging, but the authors managed to include them all in a cohesive manner.

**Weaknesses:**

1. **Presentation inconsistencies and limited analysis:**
  While the paper is generally well-written, some presentation issues remain. Figures in the experimental sections lack detailed captions, and the naming of methods is inconsistent (e.g., Table 1 refers to “GraIP” and “Max-DAG I-MLE,” which differ from names used in the text). In addition, the analysis of experimental results is minimal — sections such as causal discovery and graph rewiring offer only brief quantitative summaries without deeper interpretation or discussion. Expanding these analyses would clarify what is being learned, why specific methods succeed or fail, and the advantages of seeing these problems through the lens of inverse problems. Also, CD is mentioned in Section 4.4 (I assumed it refers to causal discovery), but it is never defined.

2. **Unclear formulation:**
  The definition of the general framework could be more precise. In Section 2.1, the forward model maps from latent variables $z$ (and noise) to observations $y$, and the inverse mapping goes from $y$ back to $z$. In the graph case, however, the inverse mapping $I(G, X, y) \to \tilde{G}$ does not strictly invert the forward map $F(G, X) \to y$; instead, it modifies or reconstructs the graph itself. This makes the notion of “inverse” somewhat ambiguous. The authors should clarify what is being inverted conceptually. In particular, whether $I$ aims to recover the underlying graph structure or optimize a reconstruction objective involving $F$.

3. **Mismatch between framework and instantiations:**
  In some applications, especially graph rewiring, the inverse mapping operates within the same space (from $G$ to $\tilde{G}$). At the same time, the forward map serves as a predictor for some downstream task (link prediction). This setup appears closer to a graph topology/graph learning or representation learning combined with a regressor than a classical inverse problem, at least from a conceptual perspective. The authors should better explain how such tasks fit within the general GrapIP framework and how using this formulation yields some improvement (even if not numerically, but at least conceptually).

4. **Missing related work:**
  The related work section omits a recent work on *graph inverse problems*  [1], which is directly relevant and should be cited for completeness.

5. **Comparison between blind and non-blind cases**
The framework supports both blind and non-blind cases, but it is not clear how going from the blind to the non-blind case complicates the general formulation. It would be good to see an experiment showing how well the proposed framework estimates a forward model, for example, by doing this in a controlled case where you have access to the forward model. This would be important to 1) understand if the method can recover the forward model, and 2) how stable the joint optimization is.

6. **Discretization bottleneck**
 Section 3.1 says, "It is sometimes preferable to relax the requirement to produce a
discrete graph during training. An important insight from our framework is that discretization can be
harmful, impairing stability and convergence, and also beneficial, by enforcing useful structural priors
early in learning". Can the authors expand on the trade-off, and why is it sometimes preferable to relax the discrete requirements?

7. **Unclear contribution of GraIP**
What are the concrete benefits of this general formulation?
Can a single model be transferred across domains/applications? Or does the framework enable new methods?
Furthermore, the paper proposes MPNN+I-MLE as a universal recipe, but results show it's often worse than continuous methods.
In particular, a core claim is that GraIP enables "cross-pollination of existing methods across graph inverse problems.
However, no experiments validate this. For example: a) Can an inverse map I^(θ) trained on NRI transfer to GRN inference (both involve temporal dynamics and complete graphs)? b) Does pre-training on one GraIP task improve performance on another?
It is important to demonstrate concrete transfer benefits or acknowledge this as a limitation and future work.

8.  **Inconsistent baseline comparisons:**
The description of the baselines needs to be improved. First, "Base" and "Rand" rewire are poorly defined. What is "Base" for example? For NRI, the comparisons are only with respect to the gradient estimator method, but not any domain-specific one or non-GraIP that performs well (even some SOTA method or close to SOTA with an open source implementation).


[1] Eliasof, M., Siddiqui, M. S. R., Schönlieb, C. B., & Haber, E. (2025, April). Learning Regularization for Graph Inverse Problems. In Proceedings of the AAAI Conference on Artificial Intelligence (Vol. 39, No. 16, pp. 16471-16479).

**Questions:**

1. **Learning of the inverse map (Eq. 1):**
   In conventional inverse problems (e.g., in imaging), the inverse mapping is typically learned by minimizing an error between the output of the inverse model (e.g., a denoiser) and the clean latent signal, sometimes using frameworks like SURE or diffusion-based self-supervised training. Here, the loss seems to minimize the discrepancy between $F(I(G, X, y))$ and the *noisy observations* $y$. Could the authors clarify the rationale and how this differs from standard inverse problem formulations?

2. **Regularization:**
   The general formulation includes a regularization term $R(I(G, X, y))$, but its instantiation is not described in the experiments. Was any explicit regularization (e.g., sparsity, or smoothness) applied in the benchmarks, or was it omitted?
Furthermore, Section 4.4 explains that the ill-posedness is stronger when the size of the graphs increases, which is expected. Is there any potential regularizer that might help to address this? Because the framework incorporates a regularizer, but as said above, it is never used

3. **Generality of GraIP**
While GraIP is a unifying framework, it would be good to explain which problems do not fit GraIP?

4. **Drop in performance in CD**
Can the authors expand on why the performance of Max-DAG I-MLE in Table 1 drops so drastically from ER2-30 to ER4-30?


---

I assign a score of 4 due to the concerns outlined above, particularly the missing experimental validation of transfer claims, ambiguities between the theoretical and generic formulation, its practical instance, and unused regularization. In particular, the most relevant points are:

1. Clarify what is being "inverted" conceptually (Weaknesses #2-3)
2. Provide preliminary transfer experiments or acknowledge as future work
   (Weakness #7)
3. Explain the CD performance collapse and potential remedies (Question #4)
4. Discuss how well the method approximates the forward operator in some controlled experiment.
5. Clarify why R(·) is never instantiated (Question #2)

I would be happy to discuss the above-mentioned points and the remaining ones.

---

> ### Author Response · Authors · 2025-11-25
> **W1**
>
> We thank the reviewer, JtZo, for their questions and valuable feedback. We aim to address the concerns raised and answer their questions satisfactorily; we are, of course, open to further discussion on any additional questions. Please note that we aim to update the PDF through the rebuttal period (and by camera ready) to incorporate all changes promised here.
>
> ---
>
> > **W1: Presentation inconsistencies and limited analysis:** While the paper is generally well-written, some presentation issues remain. Figures in the experimental sections lack detailed captions, and the naming of methods is inconsistent (e.g., Table 1 refers to “GraIP” and “Max-DAG I-MLE,” which differ from names used in the text). In addition, the analysis of experimental results is minimal — sections such as causal discovery and graph rewiring offer only brief quantitative summaries without deeper interpretation or discussion. Expanding these analyses would clarify what is being learned, why specific methods succeed or fail, and the advantages of seeing these problems through the lens of inverse problems. Also, CD is mentioned in Section 4.4 (I assumed it refers to causal discovery), but it is never defined.
>
> We appreciate and acknowledge reviewer JtZo’s comments, and will refine the writing for clarity. The implemented and planned changes to the manuscript (pertaining to this specific weakness) are as follows:
>
> - The captions have been improved.
> - Naming & general clarity of tables have been improved, particularly for CD (Table 1\) and rewiring (Table 4), which also pertains to W8.
> - CD is indeed shorthand for causal discovery, this is now made clear in the beginning of Section 4.1.
>
> We additionally agree that the current analysis is somewhat condensed, mainly due to space constraints, considering the range of instances we want to cover within the GraIP umbrella. With the additional page provided for the rebuttal/camera ready, we will provide a fuller discussion section that considers the following:
>
> - An appendix section on regularization strategies and avoiding degenerate solutions in ill-posed GraIPs, with connections to the analogous problems in classical inverse problem literature.
> - A brief discussion on the scalability and efficiency of GraIPs
> - A more in-depth discussion of discretization and gradient estimation methods, their limitations and associated challenges, and how these outline a common family of challenges of discrete GraIP methods across domains (related to Weakness \#6)
> - As future directions, a discussion of hybrid GraIPs (transferability between continuous-to-discrete GraIPs and vice versa) and potential applications to graph foundation models (GFM).
> - A more in-depth discussion of blind vs. non-blind GraIPs (related to Weakness \#5).

---

> ### Author Response · Authors · 2025-11-25
> **W2**
>
> > **W2: Unclear formulation:** The definition of the general framework could be more precise. In Section 2.1, the forward model maps from latent variables $z$ (and noise) to observations $y$, and the inverse mapping goes from $y$ back to $z$. In the graph case, however, the inverse mapping $I(G, X, y) \to \tilde{G}$ does not strictly invert the forward map $F(G, X) \to y$; instead, it modifies or reconstructs the graph itself. This makes the notion of “inverse” somewhat ambiguous. The authors should clarify what is being inverted conceptually. In particular, whether $I$ aims to recover the underlying graph structure or optimize a reconstruction objective involving $F$.
>
> We thank the reviewer for pointing out the ambiguity surrounding the notion of an "inverse" map. Our use of the term does not imply that the inverse map $I$ is a strict mathematical inverse of the forward map $F$. Instead, in line with the standard formulation of inverse problems in the natural sciences (Sec. 2.1), "inverse" refers to estimating latent causal factors that, when evaluated through a forward generative process, reproduce the observed data.
>
> Classical inverse problems, such as recovering the parameters of a physical simulator that explain measurement data, estimating initial conditions of a PDE from its later state, or inferring tissue properties in MRI, do not require computing $F^{-1}$ in a literal sense. Instead, the inverse operator searches for latent variables whose _forward evaluation_ best matches the observations. Our GraIP formulation follows this principle exactly.
>
> In GraIP, the latent object is a graph $\tilde{G}$, and the goal of $I(\theta)$ is not to algebraically invert $F$, but to propose a structure that _explains_ the observations $y$ under the forward map. This is especially relevant for graph domains, where the forward process is typically non-invertible, multiple graphs may induce identical observations, and the objective is often to recover a graph that optimizes a reconstruction or downstream task rather than a unique pre-image of $F$.
>
> We will clarify this in the revision by emphasizing that $I$ solves an optimization problem of the form
>
> $\tilde{G} = I(y) \quad$ such that $\quad F(\tilde{G}) \approx y,$
>
> consistent with the inverse-problem formulation in Eq. (1) and the GraIP objective in Eq. (2). This captures both scenarios mentioned: recovering an underlying generative graph structure and optimizing a reconstruction or downstream objective involving $F$.

---

> ### Author Response · Authors · 2025-11-25
> **W3**
>
> > **W3: Mismatch between framework and instantiations:** In some applications, especially graph rewiring, the inverse mapping operates within the same space (from $G$ to $\tilde{G}$). At the same time, the forward map serves as a predictor for some downstream task (link prediction). This setup appears closer to a graph topology/graph learning or representation learning combined with a regressor than a classical inverse problem, at least from a conceptual perspective. The authors should better explain how such tasks fit within the general GraIP framework and how using this formulation yields some improvement (even if not numerically, but at least conceptually).
>
> We thank the reviewer for raising this point. We agree that graph rewiring differs superficially from "classical" inverse problems, and we believe this is precisely where the GraIP perspective is valuable. As we explain in Sec. 3 (pp. 4-5) and in the dedicated rewiring instantiation in Sec. 4.3, GraIP does not require the latent object to have a different type than the input; the key requirement is that the inverse map proposes a _latent structure_, here a modified graph $\\tilde{G}$, that, when evaluated by the forward map $F$, produces outputs consistent with the observed data or downstream objectives.
>
> In graph rewiring, the inverse map $I(\\theta)$ indeed maps $G \\mapsto \\tilde{G}$ within the same space, but conceptually it plays the same role as in other inverse problems: it searches for the latent structural configuration that best explains task-driven observations when passed through the forward operator. The forward map in this setting is a downstream predictor, but this is fully aligned with the general GraIP formulation in Eq.\~(2): even in classical inverse problems, $F$ may itself be a learned surrogate or differentiable model used to evaluate how well a proposed latent structure explains observations.
>
> This task thus fits naturally into GraIP: the latent structure to be inferred is the “true” or “task-optimal” graph, the forward evaluation measures how well that structure supports the observed labels, and the optimization jointly adapts both $I$ and $F$ exactly as described in the blind inverse-problem formulation. We highlight this conceptual alignment explicitly in Sec. 4.3, where we state that rewiring “naturally aligns with the perspective of the inverse problem” because the objective is to find a graph $\\tilde{G}$ whose forward evaluation best matches the supervised signal.
>
> We added graph rewiring precisely because viewing it through the GraIP lens is informative: it exposes its shared structure with causal discovery, NRI, and GRN inference — all of which learn latent graphs by matching a forward model to observations. Conceptually, this framing unifies methods that have so far been studied in isolation, enables transfer of tools (e.g., discretization strategies such as I-MLE), and clarifies why techniques developed for other inverse graph tasks can be meaningfully brought to rewiring as well.
>
> We will make this connection even more explicit in the revision to ensure that the conceptual role of rewiring within GraIP is clear.

---

> ### Author Response · Authors · 2025-11-25
> **W4**
>
> > **W4: Missing related work** The related work section omits a recent work on graph inverse problems [1], which is directly relevant and should be cited for completeness.
> >
> > [1] Eliasof, M., Siddiqui, M. S. R., Schönlieb, C. B., & Haber, E. (2025, April). Learning Regularization for Graph Inverse Problems. In Proceedings of the AAAI Conference on Artificial Intelligence (Vol. 39, No. 16, pp. 16471-16479).
>
> We acknowledge this omission and thank reviewer JtZo; we will add the reference to related work. The cited work focuses on adapting classical regularization techniques from inverse problems to graph settings, making it a valuable complementary contribution to ours.

---

> ### Author Response · Authors · 2025-11-25
> **W5**
>
> > **W5: Comparison between blind and non-blind cases:** The framework supports both blind and non-blind cases, but it is not clear how going from the blind to the non-blind case complicates the general formulation. It would be good to see an experiment showing how well the proposed framework estimates a forward model, for example, by doing this in a controlled case where you have access to the forward model. This would be important to 1) understand if the method can recover the forward model, and 2) how stable the joint optimization is.
>
> We thank reviewer JtZo for the suggestion, and are happy to provide a brief comparison of blind and non-blind settings. For this, we use the NRI instance, as we have access to the forward simulator. For the NRI Springs benchmark, we see that the non-blind setting simplifies the task as expected: The non-blind NRI \+ STE setting improves on its blind variant (and achieves almost-perfect results) across all graph metrics for both $N=\\{5, 10\\}$ settings. However, do note that in Springs even the blind NRI can recover the forward model well and solve the task successfully.
>
> We therefore evaluate on the noisier *Charged* benchmark as well for $N=\\{5, 10\\}$, which results in an interesting observation: The non-blind model struggles severely with slightly above random graph metrics, whereas the the blind case is still successful (AUC: \~91\% for $N = 5$, \~80\%for $N = 10$). We note that the non-blind results fail due to the vanishing gradients caused by the simulation decoder, which prevents the inverse map to converge to the correct graph. This setting then showcases an "edge case" where learning in the blind setting proves more reliable than using a known forward map in the non-blind setting.
>
> Qualitatively, excluding the edge case in Charged, the results demonstrate that even in more complex, noisier NRI settings, recovery of the forward model is largely possible, though of course with some caveats on stability and performance compared to the non-blind case, which represents an easier problem setting.
>
> **Springs**
> | N | Method                  | MSE &darr; | Accuracy &uarr; | F1-score &uarr; | ROC-AUC &uarr; |
> |----|--------------------------|-----------------|-----------------------|----------------------|-------------------------|
> | 5 | NRI + STE             | 1.9e-4 ± 0.0           | 99.4 ± 0.3            | 99.3 ± 0.3            | 99.9 ± 0.0            |
> | 5 | NRI + STE (non-blind) | 1.7e-7 ± 0.0 | 99.8 ± 0.0         | 99.8 ± 0.0            | 100. ± 0.0               |
> | 10 | NRI + STE             | 3.3e-5 ± 0.0    | 98.4 ± 0.0            | 98.2 ± 0.0             | 74.9 ± 0.3              |
> | 10 | NRI + STE (non-blind) | 1.2e-6 ± 0.0 | 98.2 ± 0.1         | 98.2 ± 0.1            | 99.6 ± 0.0               |
>
> **Charged**
> | N | Method                  | MSE &darr; | Accuracy &uarr; | F1-score &uarr; | ROC-AUC &uarr; |
> |----|--------------------------|-----------------|-----------------------|----------------------|-------------------------|
> | 5 | NRI + STE             | 1.2e-3 ± 0.0    | 82.8 ± 0.1            | 82.5 ± 0.1             | 91.0 ± 0.8              |
> | 5 | NRI + STE (non-blind) | 1.8e-4 ± 0.0 | 52.6 ± 2.4         | 42.3 ± 7.0            | 57.6 ± 3.6               |
> | 10 | NRI + STE             | 1.6e-3 ± 0.0    | 70.8 ± 0.7            | 70.2 ± 0.8             | 80.4 ± 1.0              |
> | 10 | NRI + STE (non-blind) | 7.5e-4 ± 0.0 | 53.4 ± 2.1         | 49.3 ± 7.3            | 56.5 ± 2.7               |

---

> ### Author Response · Authors · 2025-11-25
> **W6**
>
> > **W6: Discretization bottleneck:** Section 3.1 says, "It is sometimes preferable to relax the requirement to produce a discrete graph during training. An important insight from our framework is that discretization can be harmful, impairing stability and convergence, and also beneficial, by enforcing useful structural priors early in learning". Can the authors expand on the trade-off, and why is it sometimes preferable to relax the discrete requirements?
>
> In essence, the preference for continuous relaxations for inherently discrete problems arises from the inherent biases and lack of stability that affects gradient estimation methods for discrete problems. These difficulties are well-documented and are directly connected with a fundamental trade-off between unbiased but high-variance estimators (e.g., the score-function estimator) and low-variance but biased estimators or relaxations (e.g., straight-through, Gumbel-Softmax, I-MLE) (Titsias & Shi, 2022; Minervini et al., 2023). For I-MLE in particular, Minervini et al. (2023) show that the finite-difference step size $\\lambda$ directly trades off gradient sparsity vs. bias: as $\\lambda \\rightarrow 0$, the estimated gradients become zero almost everywhere (completely uninformative), whereas larger $\\lambda$ values yield denser but increasingly biased gradients. In our GraIP instantiations this interacts with highly non-convex discrete objectives, so discretized training is prone to getting trapped in poor local optima unless the estimator is very carefully tuned.
>
> In contrast, continuous relaxations such as NoTears/GOLEM for CD or continuous rewiring for GNNs provide smoother optimization landscapes and well-behaved gradients, which explains why they often outperform discrete I-MLE variants in our experiments. We will make this discussion explicit in Section 4.4, summarizing the AIMLE analysis (sparsity-bias trade-off for IMLE gradients), and clarifying that our benchmarking goal is precisely to *expose* these limitations empirically rather than claim that discretization is universally beneficial.
>
> _Titsias, M. & Shi, J. (2022).  Double Control Variates for Gradient Estimation in Discrete Latent Variable Models . Proceedings of The 25th International Conference on Artificial Intelligence and Statistics, in Proceedings of Machine Learning Research 151:6134-6151 Available from https://proceedings.mlr.press/v151/titsias22a.html._
>
> _Pasquale Minervini, Luca Franceschi, and Mathias Niepert. 2023\. Adaptive perturbation-based gradient estimation for discrete latent variable models. In Proceedings of the Thirty-Seventh AAAI Conference on Artificial Intelligence (AAAI '23), Vol. 37\. AAAI Press, Article 1034, 9200–9208. https://doi.org/10.1609/aaai.v37i8.26103_

---

> ### Author Response · Authors · 2025-11-25
> **W8**
>
> > **W8: Inconsistent baseline comparisons:** The description of the baselines needs to be improved. First, "Base" and "Rand" rewire are poorly defined. What is "Base" for example? For NRI, the comparisons are only with respect to the gradient estimator method, but not any domain-specific one or non-GraIP that performs well (even some SOTA method or close to SOTA with an open source implementation).
>
> We thank the reviewer for bringing our attention to these points. We have significantly overhauled Table 4: “Base” (renamed “Baseline (GINE)”) refers to the baseline GINE model within the PR-MPNN framework trained on the original graphs, i.e., without any rewiring. “Rand. rewire” refers to the random rewiring setting where we use the same rewiring budget as the PR-MPNN, but rewire the graph randomly. These definitions have now been added to the caption. Please also note that we’ve extended the benchmarks significantly to include the long-range benchmarks *Peptides-func* and *Peptides-struct*, as well as the heterophilous WebKB datasets *Cornell, Texas & Wisconsin* to make the case for the utility of data-driven graph rewiring under a variety of heterophily/long-range interactions, and emphasize the wide array of problems and benchmarks that can be used to evaluate GraIPs.
>
> For NRI, we would like to clarify that the *NRI \+ STE model* we use is \*the\* domain-specific method proposed in the *NRI paper* (Kipf et al., 2018), as we describe in Section 4.2, under “Methods and empirical insights”. For the NRI \+ I-MLE method, we maintain an identical domain-specific NRI model, except changing the gradient estimation method to I-MLE from STE. We are happy to provide evaluations on other methods like conventional GNNs or LSTM-based models by the camera-ready for completeness; we would nevertheless like to point out that the original NRI paper convincingly shows how their proposed model we inherit here outperforms these baselines. We also kindly ask the reviewer for any specific methods he would like to see for the NRI instance.

---

> ### Author Response · Authors · 2025-11-25
> **Q1**
>
> > **Q1: Learning of the inverse map (Eq. 1):** In conventional inverse problems (e.g., in imaging), the inverse mapping is typically learned by minimizing an error between the output of the inverse model (e.g., a denoiser) and the clean latent signal, sometimes using frameworks like SURE or diffusion-based self-supervised training. Here, the loss seems to minimize the discrepancy between $F(I(G, X, y))$ and the noisy observations $y$. Could the authors clarify the rationale and how this differs from standard inverse problem formulations?
>
> We appreciate the reviewer’s observation and are happy to clarify the distinction. Many classical inverse problems in imaging indeed use a “direct”' inverse model. In these settings, the forward operator $F$ is typically fixed, linear, and known, so the inverse can be learned by directly regressing on $z$.
>
> Our formulation in Eq. (1), which matches standard _model-based_ inverse problems, corresponds to the other primary class of inverse problems where $z$ is not observable during training. This is the case in many scientific settings: for example, when inferring simulator parameters, initial conditions of dynamical systems, or physical properties of a medium, the latent $z$ is never available. Instead, one optimizes $I$ by ensuring that the forward evaluation $F(z)$ matches the observed data. Our GraIP setting falls precisely into this regime. In graph-structured inverse problems, the latent graph $\tilde{G}$ is not provided in training data, so the only principled way to learn $I$ is to match $F(I(y))$ to $y$, as formalized in Eq. (1) and Eq. (2). This is entirely consistent with standard approaches for non-supervised inverse problems in physics and system identification, where the inverse map is learned by minimizing discrepancies in the forward-model space rather than in the latent space.
>
> We will clarify this distinction in the revision: GraIP adopts the classical implicit or indirect learning formulation used whenever the true latent variables are unobserved. Thus, our deviation from denoising-style training is not a departure from inverse problem theory, but rather corresponds to the standard treatment used across domains where latents cannot be supervised directly.

---

> ### Author Response · Authors · 2025-11-25
> **Q2**
>
> > **Q2: Regularization:** The general formulation includes a regularization term $R(I(G, X, y))$, but its instantiation is not described in the experiments. Was any explicit regularization (e.g., sparsity, or smoothness) applied in the benchmarks, or was it omitted? Furthermore, Section 4.4 explains that the ill-posedness is stronger when the size of the graphs increases, which is expected. Is there any potential regularizer that might help to address this? Because the framework incorporates a regularizer, but as said above, it is never used.
>
> Regularization is certainly employed in most experiments conducted in our work; however, regularizers come in many forms across methods and domains, and we agree with the reviewer that it’s sometimes not obvious how regularization is integrated into a given GraIP solver in practice. We hereby provide several examples, we will also improve the writeup to clarify general regularizer usage and emphasize the presence of regularizers where appropriate:
>
> * In NRI, all methods employ a KL divergence-based regularization term to induce a prior over edge probabilities to encourage sparsity.
> * In GRN, a similar (optional) prior over edge probabilities is introduced, as per RiTINI (Bhaskar et al., 2024): Based on domain knowledge on the GRN of interest, a partial graph is used as a prior to “fix” edges that are known to exist between different genes. The GraIP method then learns to “complete” this partial graph, resulting in a more tractable (i.e., less ill-posed) problem.
> * In CD, the continuous relaxation based NoTears method employs two regularizers, one that encourages sparsity and one that promotes acyclicity.
>
> Essentially, all of these regularization techniques aim to tackle the ill-posedness we discuss in Section 4.4, but of course this is not a comprehensive list of all viable regularizers. Additionally, in the context of GraIP we’re primarily interested in regularizing the graph structure specifically; for certain tasks, node features may also need to be regularized in a setting more akin to classical inverse problems. For example, Tikhonov-like regularization (Benning & Burger, 2018; Aster, Borchers & Thurber, 2019) is used to induce sparsity over features, whereas Laplacian regularization can be used to impose a smoothness prior over node features (Eliasof et al., 2025). While we do not employ these regularizers due to our interest in structural regularization, it comes to show that a vast array of regularization strategies is available at our disposal for GraIPs, with some inherited from classical inverse problems literature.
>
> _Bhaskar, D., Magruder, D. S., Morales, M., De Brouwer, E., Venkat, A., Wenkel, F., Noonan, J., Wolf, G., Ivanova, N., and Krishnaswamy, S. Inferring dynamic regulatory interaction graphs from time series data with perturbations. In Learning on Graphs (LoG) Conference. PMLR, 2024._
>
> _Benning, M., Burger, M. Modern regularization methods for inverse problems. Acta Numerica. 2018;27:1-111. doi:10.1017/S0962492918000016_
>
> _Aster, R. C., Borchers, B., Thurber, C. H. Parameter Estimation and Inverse Problems (Third Edition), Elsevier, 2019. ISBN 9780128046517. doi:10.1016/B978-0-12-804651-7.00002-X._
>
> _Eliasof, M., Siddiqui, M. S. R., Schönlieb, C. B., & Haber, E. (2025, April). Learning Regularization for Graph Inverse Problems. In Proceedings of the AAAI Conference on Artificial Intelligence (Vol. 39, No. 16, pp. 16471-16479)._

---

> ### Author Response · Authors · 2025-11-25
> **Q4**
>
> > **Q4: Drop in performance in CD:** Can the authors expand on why the performance of Max-DAG I-MLE in Table 1 drops so drastically from ER2-30 to ER4-30?
>
> The poor performance of I-MLE with Max-DAG in ER4-30 is primarily driven by the increasing combinatorial complexity of the problem. Although both settings use ER graphs with 30 nodes, raising the expected degree from two to four substantially increases the graph density and the size of the DAG search space compatible with the data. In this regime, many distinct graphs and parameters of the forward function are local optima, making the inverse problem significantly more ill-posed. The optimization process gets stuck in poor local optima. This ill-posedness relationship is a central point of our discussion in Section 4.4, as reviewer JtZo also noted. An analogous effect appears when comparing the NRI and GRN settings: while the formulations are similar, the GRN task involves \~20x larger graphs and far fewer training examples, leading GraIP models to achieve near-perfect accuracy on NRI but only marginally above chance on GRN.
>
> For Max-DAG with I-MLE specifically, the discretization step must repeatedly solve a much harder combinatorial problem on a denser graph. As a result, its approximate gradients become noisy, optimization frequently stalls in poor local minima, and performance undergoes a sharp “phase transition” relative to continuous relaxations such as NoTears and GOLEM. Similar phase transitions are well documented in graphical model structure learning. For example, the transition in recoverability with sample size observed by Lee & Hastie (2015) and the non-identifiability phenomena reported by Bento & Montanari (2009) for Ising models. We will clarify this interpretation in the discussion section.
>
> ​​_Jason D. Lee and Trevor J. Hastie. Learning the Structure of Mixed Graphical Models. J Comput Graph Stat. 2015 January 1; 24(1): 230–253. doi:10.1080/10618600.2014.900500._
>
> _José Bento and Andrea Montanari. 2009\. Which graphical models are difficult to learn? In Proceedings of the 23rd International Conference on Neural Information Processing Systems (NIPS'09). Curran Associates Inc., Red Hook, NY, USA, 1303–1311._

---

> > ### Comment · Reviewer_JtZo · 2025-11-25
> >
> > I appreciate the authors' effort in addressing all the questions in a detailed manner. Although it remains somewhat unclear what the concrete benefits and overall utility of this general formulation are (for instance, in deriving novel methodologies), I believe the contributions outweigh this concern. Given the paper's strengths and the fact that the detailed responses addressed almost all the points, I will raise my score.

---

### Official Review · Reviewer_Ubr3 · 2025-10-31

**Soundness:** 3
**Presentation:** 3
**Contribution:** 2
**Rating:** 4
**Confidence:** 3

**Summary:**

This paper introduces GraIP (Graph Inverse Problems), a unified conceptual and benchmarking framework that reframes a wide range of graph learning tasks—including causal discovery, neural relational inference, graph rewiring, and combinatorial optimization—as inverse problems. Instead of learning on a given graph, GraIP aims to recover the latent graph structure that best explains observed data under a learnable forward model. The authors provide a general formulation, instantiate it across several domains, and propose corresponding benchmarks and baselines. They also empirically evaluate existing methods under the GraIP lens, highlighting shared challenges such as discretization, non-identifiability, and gradient estimation.

**Strengths:**

- Originality: This is the first systematic attempt to unify graph structure learning tasks under the inverse problem paradigm. The framing is conceptually novel and provides a fresh perspective on seemingly disparate tasks like rewiring, causal discovery, and relational inference. It opens up new avenues for cross-pollination between fields like combinatorial optimization, causal inference, and graph generation.

- Technical Quality: The paper is mathematically rigorous and well-grounded in the classical inverse problem literature. The authors clearly define the forward map, inverse map, distance function, and regularization, and show how existing methods (e.g., NRI, NoTears, PR-MPNN) can be reformulated within GraIP. The benchmarking protocol is comprehensive, with standardized datasets, metrics, and baselines.

- Clarity: Despite the breadth of content, the paper is well-organized and clearly written. The modular structure (inverse map, forward map, discretization, gradient estimation) makes the framework easy to understand and extend. Figures and tables effectively summarize the instantiations and results.

- Significance: GraIP has the potential to redefine how we think about graph structure learning. By exposing shared algorithmic challenges (e.g., differentiating through discrete graphs, enforcing constraints), it encourages methodological innovation and fair comparison across domains. The framework is timely, as many graph learning tasks are currently silos with little cross-domain insight.

**Weaknesses:**

- Limited Novel Algorithmic Contributions: While GraIP is conceptually elegant, the paper does not introduce new models or algorithms. It reframes existing ones and benchmarks them under a unified lens. As a result, the algorithmic novelty is low, and the empirical improvements are marginal in some domains (e.g., causal discovery).

- Scalability and Efficiency Not Discussed: The paper does not address the computational scalability of GraIP formulations. For example, in causal discovery and GRN inference, the combinatorial explosion of possible graphs is a major bottleneck. The authors acknowledge ill-posedness at scale but do not propose principled solutions (e.g., amortized inference, hierarchical priors, or latent variable relaxations).

- Discretization Remains a Bottleneck: Although the paper highlights the discretization challenge, it does not resolve it. The empirical results show that discretized methods (e.g., I-MLE) often underperform continuous relaxations (e.g., NoTears, GOLEM). A deeper analysis of when and why discretization fails, or how to improve it, is missing.

- Lack of Theoretical Analysis: While the framework is general, there is no theoretical treatment of identifiability, sample complexity, or generalization bounds under GraIP. Such an analysis would strengthen the contribution, especially for blind inverse problems where both forward and inverse maps are learned.

**Questions:**

1. Scalability: How does GraIP scale to larger graphs? Have you tested approximate inference or latent variable relaxations to handle the combinatorial explosion?

2. Theoretical Guarantees: Are there conditions under which GraIP recovers the true graph (e.g., identifiability, consistency)? Can you provide sample complexity bounds or recovery guarantees under specific generative models?

3. Discretization Improvements: What are the main failure modes of I-MLE and other gradient estimators? Have you explored hybrid methods (e.g., continuous pre-training + discrete fine-tuning) or regularization techniques to stabilize discretization?

4. Foundation Models: Do you envision pre-training a general-purpose inverse map (e.g., a graph transformer) that can be adapted to multiple GraIP tasks? How would you handle task-specific constraints (e.g., DAG, sparsity, acyclicity)?

5. Real-World Evaluation: Have you applied GraIP to real-world datasets (e.g., protein networks, brain connectivity, social graphs)? How does it perform in noisy, incomplete, or adversarial settings?

---

> ### Author Response · Authors · 2025-11-25
> **W1**
>
> We thank the reviewer, Ubr3, for their questions and valuable feedback. We aim to address the concerns raised and answer their questions satisfactorily; we are, of course, open to further discussion on any additional questions. Please note that we aim to update the PDF through the rebuttal period (and by camera ready) to incorporate all changes promised here.
>
> ---
>
> > **W1:** Limited Novel Algorithmic Contributions: While GraIP is conceptually elegant, the paper does not introduce new models or algorithms. It reframes existing ones and benchmarks them under a unified lens. As a result, the algorithmic novelty is low, and the empirical improvements are marginal in some domains (e.g., causal discovery).
>
> Our work primarily functions as a *benchmarking paper*: we develop a conceptual framework to unify existing methods and domains, identify common challenges, and facilitate benchmarking across them. Our aims and framing as a benchmarking paper are accordingly listed in the title, introduction (line 79), and conclusion (lines 474-478). The ICLR call for papers lists "datasets and benchmarks" in relevant subject areas, indicating that benchmarking papers (which generally do not introduce new methods) are appropriate for ICLR.
>
> We would like to note that while our focus is not on introducing new methods as a benchmarking paper, our paper nevertheless provides a wide range of novel insights by the aforementioned integration of standard discretization and gradient estimation methods into the pipeline, in addition to the analyses that derive from the unification of methods and benchmarks from different research communities; reviewer Ubr3 essentially acknowledges this as part of the paper’s strengths, which we appreciate.

---

> ### Author Response · Authors · 2025-11-25
> **W2 & Q1**
>
> > **W2:** Scalability and Efficiency Not Discussed: The paper does not address the computational scalability of GraIP formulations. For example, in causal discovery and GRN inference, the combinatorial explosion of possible graphs is a major bottleneck. The authors acknowledge ill-posedness at scale but do not propose principled solutions (e.g., amortized inference, hierarchical priors, or latent variable relaxations).
>
> > **Q1:** Scalability: How does GraIP scale to larger graphs? Have you tested approximate inference or latent variable relaxations to handle the combinatorial explosion?
>
> We agree that scalability is a central challenge for many GraIPs, especially in causal discovery and GRN inference, where the combinatorial growth of the graph space creates severe optimization and identifiability bottlenecks. Our goal in this work is not to propose new inference algorithms, but to introduce a common benchmarking framework that makes these difficulties visible and comparable across tasks.
>
> Approaches such as amortized inference, hierarchical priors, and latent-variable or low-rank relaxations are indeed promising ways to address large-scale ill-posedness, and we see them as natural next steps within the GraIP framework. However, incorporating them would require designing full new model families and training pipelines, which is beyond the scope of this benchmark paper. Instead, our experiments deliberately inherit the limitations of existing discretization and gradient-estimation techniques, and our large-graph CD/GRN results empirically highlight where these methods start to break down, and reinforce these known theoretical challenges (please see our response to W3 & Q3 for more on this).
>
> We will make these connections more explicit and emphasize that the limitations we observe are not unique to GraIP, but instead reflect open problems in learning with discrete graph structures more broadly. Additionally, we will add a dedicated subsection outlining principled directions for improving scalability, including amortized inference, hierarchical priors, and latent-variable relaxations. Developing these directions highlight the fact that one of the main purposes of GraIP is precisely to surface such cross-cutting challenges and stimulate progress on scalable inverse-graph solvers.

---

> ### Author Response · Authors · 2025-11-25
> **W3 & Q3**
>
> > **W3:** Discretization Remains a Bottleneck: Although the paper highlights the discretization challenge, it does not resolve it. The empirical results show that discretized methods (e.g., I-MLE) often underperform continuous relaxations (e.g., NoTears, GOLEM). A deeper analysis of when and why discretization fails, or how to improve it, is missing.
>
> > **Q3:** Discretization Improvements: What are the main failure modes of I-MLE and other gradient estimators? Have you explored hybrid methods (e.g., continuous pre-training + discrete fine-tuning) or regularization techniques to stabilize discretization?
>
> The limitations of gradient estimation methods particularly in discrete problems can be traced back to a fundamental trade-off between unbiased but high-variance estimators (e.g., the score-function estimator) and low-variance but biased estimators or relaxations (e.g., straight-through, Gumbel-Softmax, I-MLE) (Titsias & Shi, 2022; Minervini et al., 2023). For I-MLE in particular, Minervini et al. (2023) show that the finite-difference step size $\\lambda$ directly trades off gradient sparsity vs. bias: as $\\lambda \\rightarrow 0$, the estimated gradients become zero almost everywhere (completely uninformative), whereas larger $\\lambda$ values yield denser but increasingly biased gradients. In our GraIP instantiations, this interacts with highly non-convex discrete objectives, and thus discretized training is prone to getting trapped in poor local optima unless the estimator is very carefully tuned. The AIMLE paper (Minervini et al., 2023) aims to resolve this trade-off by _learning_ the trade-off between gradient sparsity and bias per task, for example; but typically any method that tries to reduce gradient bias or variance will do so at a cost towards the other, or at a higher computational cost. In contrast, continuous relaxations such as NoTears/GOLEM for CD provide smoother optimization landscapes and well-behaved gradients, which explains why they often outperform discrete I-MLE variants in our experiments.
>
> We will make this discussion explicit in Section 4.4, summarizing the AIMLE analysis (sparsity-bias trade-off for IMLE gradients), and clarifying that our benchmarking goal is precisely to *expose* these limitations empirically rather than claim that discretization is universally beneficial.
>
> _Titsias, M. & Shi, J. (2022).  Double Control Variates for Gradient Estimation in Discrete Latent Variable Models . Proceedings of The 25th International Conference on Artificial Intelligence and Statistics, in Proceedings of Machine Learning Research 151:6134-6151._
>
> _Minervini, P., Franceschi, L., and Niepert, M. 2023. Adaptive perturbation-based gradient estimation for discrete latent variable models. In Proceedings of the Thirty-Seventh AAAI Conference on AI (AAAI '23), Vol. 37. AAAI Press, Article 1034, 9200–9208._
>
> ---
>
> **Hybrid methods:** While we did not experiment with hybrid methods, this is an interesting avenue for future work, particularly for transfer learning on data-scarce settings, and thank reviewer Ubr3 for the suggestion. For example, GraIP model pre-training on large amounts of simulated or synthetic data (e.g., NRI, where the Springs data has discrete solutions, but has 10Ks of examples) can potentially help GRN tasks (where the number of data points may be in the hundreds or low-thousands). The question of hybrid training is thus also related to the question of foundation models for GraIPs, which we also address below.
>
> We will expand our discussion and future work to include hybrid models. We also briefly note that opening up such new avenues of future work is precisely what we aim for by proposing this unified framework.
>
> ---
>
> **Regularization:** Regularization is certainly employed in most experiments conducted in our work; however, regularizers come in many forms across methods and domains, how regularization is integrated into a given GraIP solver in practice is typically method-dependent. We hereby provide several examples – essentially, all of these regularization techniques aim to tackle the ill-posedness we discuss in Section 4.4, but of course this is not a comprehensive list of all viable regularizers. We will also improve the writeup to clarify general regularizer usage and emphasize the presence of regularizers where appropriate:
>
> - In NRI, all methods employ a KL divergence-based regularization term to induce a prior over edge probabilities to encourage sparsity.
> - In GRN, a similar (optional) prior over edge probabilities is introduced, as per RiTINI: Based on domain knowledge on the GRN of interest, a partial graph is used as a prior to “fix” edges that are known to exist between different genes. The GraIP method then learns to “complete” this partial graph, resulting in a more tractable (i.e., less ill-posed) problem.
> - In CD, the continuous relaxation based NoTears method employs two regularizers, one that encourages sparsity and one that promotes acyclicity.

---

> ### Author Response · Authors · 2025-11-25
> **W4 & Q2**
>
> > **W4:** Lack of Theoretical Analysis: While the framework is general, there is no theoretical treatment of identifiability, sample complexity, or generalization bounds under GraIP. Such an analysis would strengthen the contribution, especially for blind inverse problems where both forward and inverse maps are learned.
>
> > **Q2:** Theoretical Guarantees: Are there conditions under which GraIP recovers the true graph (e.g., identifiability, consistency)? Can you provide sample complexity bounds or recovery guarantees under specific generative models?
>
> We thank the reviewer for highlighting this important direction. Questions of identifiability, sample complexity, and generalization are indeed central to inverse problems, especially in the blind setting. A comprehensive theoretical treatment would further strengthen the contribution. However, such an analysis is beyond the scope of a benchmarking and unification paper of this kind; any rigorous analysis would necessarily rely on strong, task-specific assumptions about each generative model, and would mostly recover or rephrase existing identifiability results in those subfields. Our present contribution in GraIP is therefore to make these problems comparable and decomposable by isolating the individual components like inverse/forward maps, discretization and regularization so that future theoretical work can target each component explicitly. We however do provide a preliminary probing of certain theoretical directions, such as the discussions on the relationship between scale and ill-posedness as well as the role of regularization and priors in Section 4.4.
>
> That said, we fully recognize that our unified perspective brings these theoretical issues to the forefront. In fact, identifiability and sample-complexity challenges arise precisely because GraIP reveals a common inverse-problem core across tasks where such connections were previously opaque. In the revision, we will add a discussion section that more explicitly outlines these theoretical questions and situates them within existing results on discrete optimization and gradient estimators. We view this as a compelling avenue for future research, and we hope that GraIP serves as a foundation for developing a rigorous theory of inverse problems on graphs.

---

> ### Author Response · Authors · 2025-11-25
> **Q4**
>
> > **Q4:** Foundation Models: Do you envision pre-training a general-purpose inverse map (e.g., a graph transformer) that can be adapted to multiple GraIP tasks? How would you handle task-specific constraints (e.g., DAG, sparsity, acyclicity)?
>
> Indeed, pre-training a foundation model (FM) for GraIP tasks is a viable idea with high potential impact – one we do not consider within the scope of this work primarily due to the high computational demands of such an undertaking, but certainly a fantastic research idea that coincides nicely with the increased interest in graph FMs (Liu et al., 2025; Wang et a., 2025). One can also consider GraIP FMs for more constrained settings, such as causal problems, as a large subset of GraIPs (e.g., CD/GRN/NRI) operate by learning causal relationships.
>
> In GraIP FM training, handling task-specific constraints would likely be delegated to a differentiable function that operates on the graph prior – the CD setting (Section 4.1) provides a blueprint: Proposal graphs are “filtered” to remove cycles in the discretization step by the Greedy Feedback Arc Set (Eades et al., 1993). Thus, one can generate non-constrained graphs first, and enforce constraints afterwards in a differentiable manner.
>
> We will integrate this discourse on GraIP FMs in the discussion and future work sections alongside the one on hybrid models; we hope the reviewer agrees with us that the discussions we’re (gladly\!) having regarding these fruitful future directions is what our work very much intends to do: Encouraging the community to view GraIPs as a large family of methods that share common challenges and therefore think about common potential solutions.
>
> _Jiawei Liu, Cheng Yang, Zhiyuan Lu, Junze Chen, Yibo Li, Mengmei Zhang, Ting Bai, Yuan Fang, Lichao Sun, Philip S. Yu, and Chuan Shi. 2025. Graph Foundation Models: Concepts, Opportunities and Challenges. IEEE Trans. Pattern Anal. Mach. Intell. 47, 6 (June 2025), 5023–5044. [https://doi.org/10.1109/TPAMI.2025.3548729](https://doi.org/10.1109/TPAMI.2025.3548729)_
>
> _Zehong Wang, Chuxu Zhang, Jundong Li, Nitesh Chawla, and Yanfang Ye. 2025. Graph Foundation Models: Challenges, Methods, and Open Questions. In Proceedings of the 31st ACM SIGKDD Conference on Knowledge Discovery and Data Mining V.2 (KDD '25). Association for Computing Machinery, New York, NY, USA, 6184–6194. [https://doi.org/10.1145/3711896.3736568](https://doi.org/10.1145/3711896.3736568)_

---

> ### Author Response · Authors · 2025-11-25
> **Q5**
>
> > **Q5:** Real-World Evaluation: Have you applied GraIP to real-world datasets (e.g., protein networks, brain connectivity, social graphs)? How does it perform in noisy, incomplete, or adversarial settings?
>
> Having added more benchmarks, we now evaluate on several real-world-derived datasets: For graph rewiring (Section 4.3), we have added several non-molecular, heterophilic node-level tasks *Cornell, Texas & Wisconsin*, as well as the long-range graph benchmarks (LRGB, Dwivedi et al., 2022\) *Peptides-func* (multi-label graph classification) and *Peptides-struct* (multi-label graph property regression). We currently provide results only for the baseline model and PR-MPNN with SIMPLE gradient estimation; we will further update the table through the rebuttal period, and promise the complete version (with the ablation on discretization methods) by camera-ready.
>
> | Method| Cornell &uarr; | Texas &uarr; | Wisconsin &uarr; | Peptides-func &uarr; | Peptides-struct &darr; |
> | ----|---------------|--------------|---| ---|---|
> |Base | 0.574 ± 0.006 | 0.674 ± 0.010 | 0.697 ± 0.013  | 0.550 ± 0.008 | 0.355 ± 0.005 |
> |Random rewiring | 0.510 ± 0.057 | 0.738 ± 0.012 | 0.731 ± 0.005 | 0.651 ± 0.003 | 0.251 ± 0.001 |
> |Rewired (SIMPLE) | 0.659 ± 0.040 | 0.827 ± 0.032 | 0.750 ± 0.015 | 0.683 ± 0.009 | 0.248 ± 0.001 |
>
> In general, though, we aimed to leverage “pseudo-real” datasets (Glymour, Ramsey & Zhang, 2019; Brouillard et al., 2025\) when quality real-world data isn’t available. Such datasets are well-grounded in real-life phenomena while retaining the benefits of synthetic datasets such as control over parameters, and most importantly, access to a known ground-truth graph. The Springs dataset for NRI (Section 4.2) is generated via simulating Newtonian mechanics, for example, and therefore provides realistic scenarios with chaotic behavior within a controlled environment. The newly added, noisier *Charged* NRI benchmark (see results below) is similarly based on simulating charged particles:
>
> | N | Method                  | MSE &darr; | Accuracy &uarr; | F1-score &uarr; | ROC-AUC &uarr; |
> |----|--------------------------|-----------------|-----------------------|----------------------|-------------------------|
> | 5 | NRI + STE             | 1.2e-3 ± 0.0    | 82.8 ± 0.1            | 82.5 ± 0.1             | 91.0 ± 0.8              |
> | 5 | NRI + STE (non-blind) | 1.8e-4 ± 0.0 | 52.6 ± 2.4         | 42.3 ± 7.0            | 57.6 ± 3.6               |
> | 10 | NRI + STE             | 1.6e-3 ± 0.0    | 70.8 ± 0.7            | 70.2 ± 0.8             | 80.4 ± 1.0              |
> | 10 | NRI + STE (non-blind) | 7.5e-4 ± 0.0 | 53.4 ± 2.1         | 49.3 ± 7.3            | 56.5 ± 2.7               |
>
> Finally, our GRN instantiation follows the RiTINI pipeline and uses the well-known SERGIO simulator to simulate gene expression trajectories under biologically realistic regulatory networks, with parameters chosen to mimic real differentiation processes; we then fit MIOFlow and infer interaction graphs from these trajectories. For such pseudo-real datasets, while the data are simulated, the tasks themselves represent genuine real-world use cases of GraIPs.
>
> Across the two instantiations, the GraIP baselines are quite performant: For all real-world rewiring baselines, PR-MPNN rewiring improves over the non-rewired baselines, demonstrating its under a variety of heterophily/long-range interactions, and emphasize the wide array of problems and benchmarks that can be used to evaluate GraIPs. In Charged, we again see that the ground truth graph can be recovered with high accuracy even in noisy environments in a blind setting, while the non-blind variant solves the task almost perfectly.
>
> Finally, we also note that the synthetic benchmarks serve as proof-of-concept demonstrations primarily, while being viable (e.g., realistic, sufficiently large, non-saturated) benchmarks themselves. Our focus is on the structural aspects of each problem and the corresponding solution strategies, rather than on the specific origin of the data. Whether the data come from synthetic or real sources, any method that addresses the same inverse problem in a comparable manner (e.g. the proposed synthetic benchmark or a perturb-seq based biomolecular network inference task for causal discovery) can be evaluated within the GraIP framework. We however certainly agree that integrating more real-world datasets across GraIP instances will increase its potential impact, and thus constitutes an important direction for future work.
>
> *Clark Glymour, Joseph D Ramsey, and Kun Zhang. The evaluation of discovery: Models, simulation and search through “big data”. Open Philosophy, 2(1):39–48, 2019\.*
>
> *Brouillard, Philippe & Squires, Chandler & Wahl, Jonas & Kording, Konrad & Sachs, Karen & Drouin, Alexandre & Sridhar, Dhanya. (2024). The Landscape of Causal Discovery Data: Grounding Causal Discovery in Real-World Applications. 10.48550/arXiv.2412.01953.*

---

> > ### Comment · Reviewer_Ubr3 · 2025-11-27
> > **Thanks for rebuttal.**
> >
> > Thank you for the authors' reply. As my concerns have largely been clarified, I have updated the evaluation.
> >
> > Best,
> >
> > Reviewer Ubr3

---

### Official Review · Reviewer_uJ8E · 2025-11-01

**Soundness:** 2
**Presentation:** 4
**Contribution:** 2
**Rating:** 4
**Confidence:** 3

**Summary:**

The paper introduces the GraIP framework, which formalizes different graph learning tasks—such as causal discovery, neural relational inference, and graph rewiring—as neural graph inverse problems. The goal of the graph inverse problem is to find both forward and inverse map. The inverse map recovers latent graph structures from observations, which is a tuple of the initial graph, node features, and the labels.  The forward map takes the output produced by the inverse map and predicts the outcomes. They instantiate GraIP across several domains, summarize the baseline implementations under this framework, benchmark datasets, and provide empirical results, aiming to bridge isolated methods and encourage cross-domain advancements.

**Strengths:**

- The paper is well-written and structured, ideas are presented clearly and easy to follow.
- Interesting perspective on existing problems under the inverse problem lens, with detailed analysis casting them as instances of the proposed framework.
- Interesting discussion on practical challenges: discretization bottleneck and ill-posedness for large graphs, along with potential future combinations of generative models.

**Weaknesses:**

- While the unification is interesting, the core idea of framing graph inference as an inverse problem is not fundamentally new. The framework is largely a re-framing of existing approaches rather than a methodological advance.
- While integrating discretization methods like I-MLE or STE for some existing models is somewhat novel, the core framework largely reframes existing methods without introducing new technical improvements.
- Benchmarks for the first two instances are based on synthetic data, which is limited in scale and diversity, and restricted to ideal scenarios.
- Limited baselines for each set of problems examined.

**Questions:**

- While showing existing methods follow the proposed framework, does it lead to any new insights about these methods' limitations or common failure modes that weren't apparent when they were considered in isolation?
- In the setting where both inverse and forward maps are learned, the system could find degenerate solutions (e.g., $F$ learns to ignore $\tilde{G}$ , or $I$ produces a trivial graph that $F$ easily memorizes). How can we check and guard against this and ensure the inferred graph $\tilde{G}$ is meaningful?

---

> ### Author Response · Authors · 2025-11-25
> **W1**
>
> We thank the reviewer, uJ8E, for their questions and valuable feedback. We aim to address the concerns raised and answer their questions satisfactorily; we are, of course, open to further discussion on any additional questions. Please note that we aim to update the PDF through the rebuttal period (and by camera ready) to incorporate all changes promised here.
>
> ---
>
> > **W1:** While the unification is interesting, the core idea of framing graph inference as an inverse problem is not fundamentally new. The framework is largely a re-framing of existing approaches rather than a methodological advance.
>
> On the contrary, we argue that *the explicit framing of* graph inference as an inverse problem is, in fact, fundamentally new. There is minimal prior work on connecting inverse problems with graph structures, the most relevant of which is Eliasof et al. (2025), which was brought to our attention by reviewer JtZo; all other works (Yan, Fang, and He (2023); Huang
> et al. (2023); Ling et al. (2024)) are concerned explicitly with *source estimation problems* over known graphs, and lack any general formulation beyond the problem interest and are not related with structural inference. Eliasof et al. (2025) is relevant in that it goes beyond source estimation to provide a general framework, but for inverse problems defined over the *feature* space only; they are thus not interested in *structural* inverse problems.
>
> The kind of inverse problem that these papers tackle is fundamentally different from GraIPs. In that setting, one is interested in predicting node/edge features or properties from *known* graph structures (source estimation, graph transport, etc.). In that sense, solutions to both their forward and inverse problems are defined over node or edge *features* rather than graph structures. To follow up on the source estimation example, the forward map in this case is a $k$-step diffusion process defined by the transition matrix $\\mathbf{P}^k$, and the goal is to identify the source node from the final node features after diffusion. The inverted process relies on the known graph structure, but does not learn or optimize it, indicating a distinct inverse problem paradigm than the one we are interested in.
>
> In short, though all the tasks we consider have been tackled by the literature in a domain-specific context, our contributions in providing a general unified framework for inverse problems, as well as the overview of everyday challenges, solution strategies, and insights we synthesize in light of this unified framework, are novel to the best of our knowledge. We will update our related work section to reflect the discussion above. We also kindly ask reviewer uJ8E to reference any works we may not be aware of in this context, and we would be happy to evaluate and consider them accordingly.
>
> _Eliasof, M., Siddiqui, M. S. R., Schönlieb, C. B., & Haber, E. (2025, April). Learning Regularization for Graph Inverse Problems. In Proceedings of the AAAI Conference on Artificial Intelligence (Vol. 39, No. 16, pp. 16471-16479)._

---

> ### Author Response · Authors · 2025-11-25
> **W2**
>
> > **W2:** While integrating discretization methods like I-MLE or STE for some existing models is somewhat novel, the core framework largely reframes existing methods without introducing new technical improvements.
>
> As stated in the paper, GraIP is intended as a conceptual framework rather than a source of new technical advances. Its primary purpose is to support systematic benchmarking. Our aims and framing as a benchmarking paper are accordingly listed in the title, introduction (line 79) and conclusion (lines 474-478). Benchmarking papers generally do not introduce new methods, and the ICLR call for papers lists "datasets and benchmarks" in relevant subject areas, confirming that such work is appropriate for the venue.
>
> Beyond the framework itself, our paper also contributes new insights by:
> - Integrating standard discretization methods into the pipeline and analyzing their comparative performance, a distinct methodological contribution.
> - Bringing together concepts from inverse problems, discrete gradient estimators and diverse problem settings in a novel manner.
> - Analyzing common challenges across GraIP instances in depth, and accordingly identifying directions for future work.

---

> ### Author Response · Authors · 2025-11-25
> **W3**
>
> > **W3:** Benchmarks for the first two instances are based on synthetic data, which is limited in scale and diversity, and restricted to ideal scenarios.
>
> The synthetic benchmarks primarily serve as proof-of-concept demonstrations, while also being viable benchmarks themselves (e.g., sufficiently large and non-saturated). Our focus is on the structural aspects of each problem and the corresponding solution strategies, rather than on the specific origin of the data. Whether the data come from synthetic or real sources, any method that addresses the same inverse problem comparably can be evaluated within the GraIP framework.
>
> To drive home this point, let us focus on the *Springs* NRI benchmark: We chose the Springs benchmark primarily because of its intuitiveness and maintaining a realistic setting. While the generated examples are synthetic, the underlying dynamics are physics-informed in that they follow Newtonian mechanics, so the generated examples are nevertheless representative of real-life data. More importantly, our primary goal goes beyond the Springs benchmark to demonstrate the *NRI problem itself* as a GraIP. The proposed benchmark and baseline models employ the same technical elements as the original NRI paper (Kipf et al., 2018) and accurately replicate the Springs results; thus, they serve as a case study demonstrating that *any* NRI benchmark dataset and model fit the GraIP framework. Kipf et al. (2018) accordingly propose additional synthetic physics-driven datasets (*Charged*, *Kuramoto*) as well as real-world-derived datasets (*Motion capture*, *pick-and-roll NBA data*). We argue that our empirical results on Springs are therefore sufficient to convey that all other NRI settings employed in Kipf et al. (2018) solve GraIPs themselves.
>
> In general, though, we aimed to leverage “pseudo-real” datasets (Glymour, Ramsey & Zhang, 2019; Brouillard et al., 2025) when quality real-world data isn’t available. Pseudo-real datasets are well-grounded in real-life phenomena while retaining the benefits of synthetic datasets such as control over parameters, and most importantly, access to a known ground-truth graph. The Springs dataset for NRI and our GRN data are such examples for which the data are simulated, but the tasks themselves are genuine real-world use cases of GraIPs.
>
> Finally, please note that we have added several new more real-world-derived benchmarks for data-driven rewiring (long-range graph benchmark datasets Peptides-func/struct, WebKB-based hyperlink datasets Cornell, Texas & Wisconsin), and a new physics-driven pseudo-real NRI benchmark (charged particle simulation *Charged*).

---

> ### Author Response · Authors · 2025-11-25
> **W4**
>
> > **W4:** Limited baselines for each set of problems examined.
>
> In our benchmarking setup, we prioritized casting a wide net over the *types* of problems (i.e., instantiations) that constitute GraIPs over the number of baseline models considered per problem. This is driven by the fact that our primary contribution is to demonstrate the wide range of graph learning problems that effectively implement GraIPs, a goal that goes beyond any specific benchmark dataset or baseline model considered in our evaluation. The individual benchmark datasets and models we considered in our experiments serve as a case study to demonstrate that *any* dataset-model pair that conforms to the problem setting provided fits the GraIP framework.
>
> Nevertheless, we strive to cover a range of models for each benchmark and avoid trivial evaluations. Please also note that, as mentioned in Section 4 (lines 263-265), we cover two additional example instantiations, combinatorial optimization (CO) and gene regulatory networks (GRN), in Appendix B, due to page limits.
>
> - In CO, four GNN-based GraIP baselines are evaluated on the maximum independent set (MIS) and maximum clique problems, with an ablation across non-discretized and discretized training settings, yielding 32 additional baseline evaluations.
> - In GRN, we evaluate two baseline models (GAT and graph transformer-based) on short-horizon and long-horizon gene expression prediction tasks, providing four additional baseline evaluations.
>
> Summary tables for all problem settings (12 benchmark tasks across five problem instances in total, not counting variants of the same benchmarks within CD and GRN as distinct) and baseline GraIP methods (22 in total) are available in Appendix E.
>
> On a final note, we agree with reviewer uJ8E that there is always room for improvement in benchmarking works, and more relevant baseline models are welcome. Therefore, we will aim to provide additional baseline results across tasks through the rebuttal period and by the camera-ready deadline. We also kindly ask the reviewer for any specific methods or families of models he would like to see for the GraIP instances presented.

---

> ### Author Response · Authors · 2025-11-25
> **Q1**
>
> > **Q1:** While showing existing methods follow the proposed framework, does it lead to any new insights about these methods' limitations or common failure modes that weren't apparent when they were considered in isolation?
>
> We very much appreciate this comment, as it helps us build a fuller discussion section – the current analysis section where we cover such surprising outcomes or novel ideas is somewhat condensed, mainly due to space constraints, considering the range of instances we want to cover within the GraIP umbrella. We hereby provide several points on limitations and failure modes of the models evaluated; we will integrate these points into the discussion section of the paper as well:
>
> - An interesting observation on discretization \+ gradient estimation setups is that performance decrease in edge recovery in GraIPs seem to follow a “phase transition” across multiple settings. To elaborate, it *is* expected that solving GraIPs is harder on larger and/or denser graphs due to the increasing combinatorial complexity of the problem. However, what we did not expect was that in most problems, edge recovery is viable until a certain noise threshold, where recoverability almost completely collapses, leading to approximately random performance. We observe this phenomenon when comparing similar tasks on smaller NRI datasets vs larger and data-sparse GRN settings, as well as when comparing CD problems of varying graph densities.
> - This “phase transition” phenomenon then leads us to the limitations of discretization and gradient estimation methods: We observe that solving GraIPs in a discrete manner typically exacerbates these recoverability issues on large, complex systems due to bias or variance induced by the estimated gradients, explaining the superior performance of continuous relaxations on some settings such as CD. These observations also allow us to draw parallels between prior work on graphical model structure learning, such as the transition in recoverability with sample size observed by Lee & Hastie (2015) and the non-identifiability phenomena reported by Bento & Montanari (2009) for Ising models. In light of this, we believe GraIP as a unifying theoretical framework makes a strong case (as well as a valuable testbed) for future work on gradient estimation methods.
> - Finally, we were surprised to see that discretization \+ gradient estimation as the *de facto* standard method family for GraIPs across instances, despite the emergence of other strategies in recent literature, e.g., discrete diffusion and graph generative modeling in general. Developing graph generative models that can be conditioned on desired graph priors would make a valuable contribution to GraIPs, and we hope this paper motivates future work towards this direction.
>
> _Jason D. Lee and Trevor J. Hastie. Learning the Structure of Mixed Graphical Models. J Comput Graph Stat. 2015 January 1; 24(1): 230–253. doi:10.1080/10618600.2014.900500._
>
> _José Bento and Andrea Montanari. 2009\. Which graphical models are difficult to learn? In Proceedings of the 23rd International Conference on Neural Information Processing Systems (NIPS'09). Curran Associates Inc., Red Hook, NY, USA, 1303–1311._

---

> ### Author Response · Authors · 2025-11-25
> **Q2**
>
> > **Q2:** In the setting where both inverse and forward maps are learned, the system could find degenerate solutions (e.g., $F$ learns to ignore $\tilde{G}$, or $I$ produces a trivial graph that $F$ easily memorizes). How can we check and guard against this and ensure the inferred graph $\tilde{G}$ is meaningful?
>
> We thank the reviewer for an insightful question that allows us to delve deeper into classical inverse problem literature, and provide a comparative analysis of how GraIP differs from prior work in terms of regularization and avoiding degenerate solutions in general. We plan to add the resulting discussion into the manuscript (likely as an appendix due to space constraints).
>
> Indeed, avoiding degenerate solutions is one of the key considerations in solving GraIPs, and inverse problems in general. In this paper, we discuss this problem in the context of ill-posedness and regularization. Ill-posedness is a more general term that encompasses inverse problems with no solution or multiple solutions; degenerate solutions arise in this context as a trivial solution among multiple ones. An extensive survey on ill-posed inverse problems is provided by Kabanikhin (2008).
>
> Avoiding degeneracy and ill-posedness in inverse problems is typically achieved via regularization techniques. In general inverse problems, there is a vast established literature on regularization, such as Tikhonov regularization (somewhat analogous to weight decay strategies in deep learning), smoothness-based regularizers, and iterative methods. We refer to Benning & Burger (2018) and Aster, Borchers & Thurber (2019) for a comprehensive overview; [Martin Burger’s lecture notes on Inverse Problems](https://www.uni-muenster.de/AMM/num/Vorlesungen/IP_WS07/skript.pdf) are also an excellent gentle introduction.
>
> Such inverse problem regularization techniques, however, are distinct from those for GraIPs, as the solutions to general inverse problems are not structured and are typically represented as vectors or matrices. Eliasof et al. (2025) then focus on learnable regularizers for inverse problems on graph data, where they also explore extensions of classical regularizers. To enforce smoothness over the graph, for example, the graph Laplacian can be used:
>
> \\\[ \\mathbf{R}(\\mathbf{x}) \= \\frac{1}{2} \\mathbf{x}^\\top \\mathbf{L} \\mathbf{x}\\\]
>
> If $\\mathbf{L}$ is replaced by the identity $\\mathbf{I}$, this is equivalent to Tikhonov regularization, where we simply aim to reduce the norm of the features. Note that in the inverse problems considered by Eliasof et al. (2025), the inverse map predicts node features rather than graph structures, indicating a setting more akin to general IPs and distinct from GraIPs.
>
> The regularization strategies we may consider in GraIPs are, on the other hand, inherently structural, and thus involve regularizing the graph itself by choosing an appropriate (e.g., sparse) prior distribution for the graph, or introducing a loss term that penalizes the number of edges in the adjacency matrix, for example. In one can also leverage domain-specific information: In RiTINI (Bhaskar et al., 2024), for example, the authors enforce meaningful graphs for gene regulatory network (GRN) inference by using a partial graph derived from domain knowledge of the GRN in question, the task of the inverse problem then becomes “completing” this partial graph rather than proposing one from scratch, alleviating the graph identifiability problem significantly.
>
> _Kabanikhin, S. I.. "Definitions and examples of inverse and ill-posed problems" Journal of Inverse and Ill-posed Problems, vol. 16, no. 4, 2008, pp. 317-357. [https://doi.org/10.1515/JIIP.2008.019](https://doi.org/10.1515/JIIP.2008.019)_
>
> _Benning, M., Burger, M. Modern regularization methods for inverse problems. Acta Numerica. 2018;27:1-111. doi:10.1017/S0962492918000016_
>
> _Aster, R. C., Borchers, B., Thurber, C. H. Parameter Estimation and Inverse Problems (Third Edition), Elsevier, 2019\. ISBN 9780128046517\. doi:10.1016/B978-0-12-804651-7.00002-X._
>
> _Eliasof, M., Siddiqui, M. S. R., Schönlieb, C. B., & Haber, E. (2025, April). Learning Regularization for Graph Inverse Problems. In Proceedings of the AAAI Conference on Artificial Intelligence (Vol. 39, No. 16, pp. 16471-16479)._
>
> _Bhaskar, D., Magruder, D. S., Morales, M., De Brouwer, E., Venkat, A., Wenkel, F., Noonan, J., Wolf, G., Ivanova, N., and Krishnaswamy, S. Inferring dynamic regulatory interaction graphs from time series data with perturbations. In Learning on Graphs (LoG) Conference. PMLR, 2024._

---

### Official Review · Reviewer_CBpu · 2025-11-01

**Soundness:** 3
**Presentation:** 3
**Contribution:** 3
**Rating:** 4
**Confidence:** 2

**Summary:**

The paper introduces \textbf{GraIP}, a unifying framework that casts diverse graph-learning tasks---including causal discovery (CD), neural relational inference (NRI), and data-driven \emph{rewiring}---as \emph{graph inverse problems}. The setup learns (i) an \emph{inverse map} $I_\theta$ that proposes a graph $\tilde G$ from observations, and (ii) a \emph{forward map} $F_\phi$ that uses $\tilde G$ to reconstruct the observations. Training minimizes a reconstruction loss (with optional regularization and discretization). The paper provides recipes for $I_\theta$ (scores/priors; Gumbel, I-MLE, STE for discretization) and $F_\phi$ (MPNN/Transformer/simulator), and benchmarks three domains (CD, NRI, rewiring), highlighting when discretization helps or hurts and how scale induces ill-posedness.

**Strengths:**

Unifying lens that cleanly separates I (inverse) vs F (forward) and makes many prior works “click together.” Easy to port methods across domains
    Benchmarking clarity: clearly specified domains, data, metrics, and simple, reproducible baselines; tables communicate the discretization story well.

**Weaknesses:**

Scope of benchmarks is modest. For CD, only linear-Gaussian synthetic settings are shown.  For NRI, only Springs is used. For rewiring, only ZINC.

The current baseline set is too narrow for a benchmarking paper. It does not fully establish where GraIP stands relative to strong alternatives across domains.

**Questions:**

Rewiring on non-molecular data: Do the gains persist on heterophilous graphs or long-range tasks?
In blind GraIPs (jointly learning F and I), how does the method train these two components?

---

> ### Author Response · Authors · 2025-11-25
> **W1 & Q1**
>
> We thank the reviewer, CBpu, for their questions and valuable feedback. We aim to address the concerns raised and answer their questions satisfactorily; we are, of course, open to further discussion on any additional questions. Please note that we will update the PDF through the rebuttal period (and by camera ready) to incorporate all changes promised here.
>
> ---
>
> > **W1:** Scope of benchmarks is modest. For CD, only linear-Gaussian synthetic settings are shown. For NRI, only Springs is used. For rewiring, only ZINC.
>
> > **Q1:** Rewiring on non-molecular data: Do the gains persist on heterophilous graphs or long-range tasks?
>
> We respectfully argue that the scope of our benchmarking framework is quite wide, particularly regarding the _types_ of inverse problems considered. Note that as mentioned in Section 4 (lines 263-265), we cover two additional example instantiations, combinatorial optimization (CO) and gene regulatory networks (GRN) in Appendix B, due to page limits.
> - In CO, four different GNN-based GraIP baselines are evaluated over the maximum independent set (MIS) and maximum clique problems with an ablation over non-discretized and discretized training settings, thus providing 32 additional baseline evaluations.
> - In GRN, we evaluate two baseline models (GAT and graph transformer-based) on  short-horizon and long-horizon gene expression prediction task, providing four additional baseline evaluations.
> Please also see Appendix E for summary tables for all problem settings (12 benchmark tasks across five problem instances in total, not counting variants of the same benchmarks within CD and GRN as distinct) and baseline GraIP methods (22 in total).
>
> However, we understand reviewer CBpu’s concerns regarding the NRI and graph rewiring, which evaluate on a single baseline dataset each. We therefore provide additional results for both: For NRI, we add results for the Charged benchmark, which represents a noisier, more difficult setting than Springs. The table below shows the NRI Charged results on $N=\{5, 10\}$:
>
> | N | Method                  | MSE &darr; | Accuracy &uarr; | F1-score &uarr; | ROC-AUC &uarr; |
> |----|--------------------------|-----------------|-----------------------|----------------------|-------------------------|
> | 5 | NRI + STE             | 1.2e-3 ± 0.0    | 82.8 ± 0.1            | 82.5 ± 0.1             | 91.0 ± 0.8              |
> | 5 | NRI + STE (non-blind) | 1.8e-4 ± 0.0 | 52.6 ± 2.4         | 42.3 ± 7.0            | 57.6 ± 3.6               |
> | 10 | NRI + STE             | 1.6e-3 ± 0.0    | 70.8 ± 0.7            | 70.2 ± 0.8             | 80.4 ± 1.0              |
> | 10 | NRI + STE (non-blind) | 7.5e-4 ± 0.0 | 53.4 ± 2.1         | 49.3 ± 7.3            | 56.5 ± 2.7               |
>
> For graph rewiring, we extend Table 3 and evaluate on several non-molecular, heterophilic node-level tasks Cornell, Texas & Wisconsin, as well as the long-range graph benchmarks (LRGB, Dwivedi et al., 2022) Peptides-func (multi-label graph classification) and Peptides-struct (multi-label graph property regression). We currently provide results only for the baseline model, random rewiring and PR-MPNN with SIMPLE gradient estimation; we promise the complete version (with the ablation on discretization methods) by camera-ready. Nevertheless, the current version is sufficient to make the case for the utility of data-driven graph rewiring under a variety of heterophily/long-range interactions, and emphasize the wide array of problems and benchmarks that can be used to evaluate GraIPs.
>
> | Method| Cornell &uarr; | Texas &uarr; | Wisconsin &uarr; | Peptides-func &uarr; | Peptides-struct &darr; |
> | ----|---------------|--------------|---| ---|---|
> |Base | 0.574 ± 0.006 | 0.674 ± 0.010 | 0.697 ± 0.013  | 0.550 ± 0.008 | 0.355 ± 0.005 |
> |Random rewiring | 0.510±0.057 | 0.738 ± 0.012 | 0.731 ± 0.005 | 0.651 ± 0.003 | 0.251 ± 0.001 |
> |Rewired (SIMPLE) | 0.659 ± 0.040 | 0.827 ± 0.032 | 0.750 ± 0.015 | 0.683 ± 0.009 | 0.248 ± 0.001 |
>
> _Vijay Prakash Dwivedi, Ladislav Rampášek, Mikhail Galkin, Ali Parviz, Guy Wolf, Anh Tuan Luu, and Dominique Beaini. 2022. Long range graph benchmark. In Proceedings of the 36th International Conference on Neural Information Processing Systems (NIPS '22). Curran Associates Inc., Red Hook, NY, USA, Article 1622, 22326–22340._

---

> ### Author Response · Authors · 2025-11-25
> **Q2**
>
> > **Q2:** In blind GraIPs (jointly learning F and I), how does the method train these two components?
>
> The joint training procedure is covered in Section 3, specifically lines 208-217, with the formulation presented in Equation 2. As per the GraIP formulation, we maintain that the system $F^{(\varphi)}(I^{(\theta)}(\cdot))$ needs to be end-to-end differentiable; thus the (differentiable) simulator $F$ in the non-blind case is replaced with a parametrized function $F^{(\varphi)}$. Since the forward map needs to operate on a graph output by the inverse map, GNNs and/or graph transformers are suitable candidates to model $F^{(\varphi)}$.
>
> Let us provide a concrete demonstration of joint training based on the neural relational inference (NRI) example (Section 4.2), where the blind GraIP problem is modeled within a variational autoencoder (VAE) framework (Kingma & Welling, 2013). The NRI inverse map $I^{(\theta)}$ consists of the VAE encoder $q_\theta$, which learns a probability distribution over the edges, from which an interaction graph $\tilde{G}$ is sampled. The GraIP forward map $F^{(\varphi)}$ is the VAE decoder $p_\varphi$, which predicts future states for each node by message-passing over $\tilde{G}$. The loss is then computed over the predicted and labeled future states (plus regularization), and is backpropagated end-to-end, i.e., through both the forward and inverse maps. The only potential issue in this training framework with a gradient-descent-based training scheme is obtaining gradients during the sampling step, which is handled using gradient estimation methods. We discuss gradient estimation strategies in Section 3.1 (lines 234-249). Also, see Section 2.4 in the original VAE paper (Kingma & Welling, 2013) for additional details regarding the “reparametrization trick” for gradient estimation. We will review our manuscript to improve the description of our joint learning pipeline and clarify any ambiguities; please feel free to point out any specific sentences/sections you would like to be addressed.
>
> We are also happy to provide a brief comparison of blind and non-blind settings. For this, we use the NRI instance, as we have access to the forward simulator. For the NRI Springs benchmark, we see that the non-blind setting simplifies the task as expected: The non-blind NRI + STE setting improves on its blind variant (and achieves almost-perfect results) across all graph metrics for both $N=5$ and $10$ settings. However, do note that in Springs even the blind NRI can recover the forward model well and solve the task successfully. We therefore evaluate on the noisier _Charged_ benchmark as well. On the 5-object Charged setting, the non-blind case again solves the task almost completely, and while the the blind case is still successful (\~81\% Accuracy/F1, \~91\% ROC-AUC), the gap to the blind setting is more significant compared to Springs. Qualitatively, the results demonstrate that even in more complex, noisier NRI settings, recovery of the forward model is largely possible, though of course with some caveats on stability and performance compared to the non-blind case which represents an easier problem setting.
>
> _Kingma, D. P., & Welling, M. (2014). Auto-Encoding Variational Bayes. In Conference proceedings: papers accepted to the International Conference on Learning Representations (ICLR) 2014 ArXiv. http://arxiv.org/abs/1312.6114_

---

> > ### Comment · Reviewer_CBpu · 2025-11-25
> >
> > Thanks for the response. I still maintain my original assessment that the benchmark is of limited scope and will keep my score.

---

> ### Author Response · Authors · 2025-11-25
> **W2**
>
> > **W2:**  The current baseline set is too narrow for a benchmarking paper. It does not fully establish where GraIP stands relative to strong alternatives across domains.
>
> Let us nevertheless answer this weakness (and the more recent reiteration of limited scope) for the sake of due diligence. We would like to rebut these points from two distinct angles.
>
> **The first angle is that quantitatively, we provide plenty of evaluations across many (a) GraIP instances, (b) benchmark datasets, and (c) model architectures.** In our benchmarking setup, we prioritized casting a wide net over the *types* of problems (i.e., instantiations) that constitute GraIPs over the number of baseline models considered per problem. This is driven by the fact that our primary contribution is to demonstrate the wide range of graph learning problems that effectively implement GraIPs, a goal that goes beyond any specific benchmark dataset or baseline model considered in our evaluation. The individual benchmark datasets and models we considered in our experiments serve as a case study to demonstrate that *any* dataset-model pair that conforms to the problem setting provided fits the GraIP framework.
>
> Nevertheless, we strive to cover a range of models for each benchmark while avoiding trivial evaluations. Also note that as mentioned in our response to **W1 & Q1**, with the two additional example instantiations CO & GRN in the appendix and the new benchmarks added to NRI & graph rewiring, we evaluate a total of 12 benchmark datasets (not counting variants of the same benchmarks within CD and GRN as distinct) across five GraIP instances, and evaluating 22 models in total. We again refer the reviewer to Appendix E for a complete list.
>
> We also emphasize that many of the methods evaluated (NRI-GNN for NRI, PR-MPNN for data-driven rewiring, NoTears & GOLEM in CD, GCON for CO) are considered at the very least competitive baselines (with some around the state-of-the-art) in their respective problem settings. We therefore find the reviewer’s claims on narrow baselines and limited scope largely unjustified, particularly after the additional results provided.
>
> Regardless, we agree with reviewer CBpu that there is always room for improvement in benchmarking works, and more relevant baseline models are welcome. Therefore, we will strive to provide additional baseline results across tasks through the rebuttal period and by the camera-ready deadline.
>
> **The second angle is that evaluating our work purely as a collation of baseline results across benchmarks misses the point, ignoring the contribution brought from the conceptual/theoretical unification and the analyses that derive from this unification.**  As stated in the paper, GraIP is intended as a conceptual framework rather than a source of new technical advances. Its primary purpose is to *support* systematic benchmarking by demonstrating how a large array of graph learning problems and accompanying solutions can be compared and decomposed by isolating common individual components (such as inverse/forward maps, discretization and regularization) and associated common challenges, and accordingly evaluated under a single umbrella. While we of course demonstrate a range of baselines for each problem instance, we do not envision our paper to function as a sort of “leaderboard” on GraIP models, and are much more interested in using these benchmark datasets and baselines to steer future research towards understanding and developing models to solve GraIPs, and provide new insights.
>
> Accordingly, beyond the framework itself, our paper also contributes such new insights by:
> \- Integrating standard discretization methods into the pipeline and analyzing their comparative performance, a distinct methodological contribution.
> \- Bringing together concepts from inverse problems, discrete gradient estimators and diverse problem settings in a novel manner.
> \- Analyzing common challenges across GraIP instances in depth, and accordingly identifying directions for future work.
>
> We feel that this dimension of our work which constitutes our main contribution as stated across the paper has been largely ignored in this review, which we would like to emphasize.

---

### Author Response · Authors · 2025-12-03
**Updated manuscript & list of changes**

We sincerely thank all reviewers for their thoughtful feedback and questions. We have addressed a large majority of the concerns raised by the reviewers in our rebuttal, and have now integrated the changes promised into the manuscript. The updated PDF highlights all changes in red for easy reference. The few experimental results and writeups that have not made it to this version due to time constraints will be included in the camera-ready version.

We hereby list the changes made to the manuscript, and outline their relation to any weaknesses or questions listed by the reviewers (and any changes we have promised in our responses accordingly):

* **Section 2.1 \[Inverse problems\]:** Additional clarification on the unavailability of latents $\\mathbf{z}$ in *implicit* inverse problems. \[Reviewer JtZo, Q1\]
* **Section 2.2 \[Related work\]:** Reference added to Eliasof et al. (2025), a relevant prior work on regularization for graph inverse problems. \[Reviewer JtZo, W4\]
* **Section 4 \[GraIP Instantiations\]:** References to specific regularization strategies used made more explicit across GraIP instances throughout the section. \[Reviewer uJ8E, Q2; Reviewer Ubr3, Q3; Reviewer JtZo, Q2\]
* **Section 4.1 \[CD\]:** Naming consistency fixed for Max-DAG I-MLE in CD. \[Reviewer JtZo, W1\]
* **Section 4.2 \[NRI\]:** Experiments on the additional *Charged* benchmark added for NRI, including analysis of the results. NRI \+ I-MLE experiments for Charged will be added for the camera-ready. \[Reviewer CBpu, W1, W2 & Q1; Reviewer uJ8E, W3; Reviewer Ubr3, Q5\]
* **Section 4.2 \[NRI\]:** Results for a *non-blind* GraIP baseline added for all Springs and Charged settings. Additional comparison and discussion of blind and non-blind settings provided under “methods and empirical insights”. \[Reviewer CBpu, Q2; Reviewer JtZo, W1 & W5\]
* **Section 4.3 \[Rewiring\]:** Additional discussion on framing data-driven rewiring as GraIP, and its unique characteristics as a structural learning task added. \[Reviewer JtZo, W3\]
* **Section 4.3 \[Rewiring\]:** Five new benchmark datasets added, all derived from real-world data: Heterophilic node-level benchmarks *Cornell*, *Texas* & *Wisconsin*, as well as the long-range graph benchmarks (LRGB, Dwivedi et al., 2022\) *Peptides-func* (multi-label graph classification) and *Peptides-struct* (multi-label graph property regression). Accompanying discussion on real-world, long-range and heterophilic benchmarks provided. Additional results from Gumbel & I-MLE will be provided by camera-ready. \[Reviewer CBpu, W1 & Q1; Reviewer uJ8E, W3; Reviewer Ubr3, Q5; Reviewer JtZo, W8\]
* **Section 4.4 \[Discussion\]:** A much more comprehensive discussion of ill-posedness and its relation with discretization and gradient estimation added. The additional discussion also aims to identify and analyze the failure cases of discretizers/gradient estimators, and GraIPs in general, serving as a summary of common challenges of GraIPs. \[Reviewer uJ8E, Q1; Reviewer Ubr3 W3 & Q3; Reviewer JtZo, W6 & Q4\]
* **Section 5 \[Conclusion & future work\]:** A brief discussion on additional future work, including hybrid training, generalization, transferability and extensions to foundation models provided. \[Reviewer Ubr3 Q3 & Q4\]
* **Appendix B.2 \[GRN\]:** References to regularization used in GRN inference clarified. \[Reviewer uJ8E, Q2; Reviewer Ubr3, Q3; Reviewer JtZo, Q2\]
* **Appendix D \[Regularization for GraIPs\]:** A new section focusing on regularization for classical inverse problems, extensions to IPs on graphs, and finally learnable regularizers provided. Existing works on regularization for IPs compared and contrasted with our work. \[Reviewer uJ8E, Q2; Reviewer JtZo, W1 & Q2\]
* **Appendix E (previously Appendix D) \[Summary tables\]:** Summary tables updated to reflect latest instances and baseline models.

---

### Author Response · Authors · 2025-12-03
**Summary of reviewer-author discussions**

In addition to the updated manuscript and the list of changes implemented (posted below), we would like to provide a brief summary of the state of the reviewer-author discussion prior to the changes made to the rebuttal process.

We have thoroughly addressed the weaknesses and questions raised by the reviewers.
- Reviewers JtZo and Ubr3 acknowledged our clarifications, additional experiments and improvements to the manuscript, and both raised their scores to 6.
- Reviewer CBpu initially maintained their score due to remaining concerns of limited scope and baselines; we subsequently addressed these concerns with additional rebuttal (titled **W2**), to which reviewer CBpu had not responded by the time of the ICLR response to the leak.
- Similarly, reviewer uJ8E had not responded to our rebuttal by the ICLR response, though we believe to have addressed all their concerns.

We are concerned that reviewers CBpu and uJ8E may not have had the opportunity to read and respond to our full set of comments before the changes to the rebuttal process were implemented. This is particularly unfortunate in the case of reviewer uJ8E, from whom we have not heard since the initial review. While we cannot be certain, the pattern exhibited by two of the four reviewers suggests that the remaining reviewers might likewise have reconsidered their assessments had they been able to further engage with our responses.

We would kindly ask the AC and SAC to take this context into account when making the final decision.

---

### Meta-Review · Area_Chair_JPkJ · 2026-01-06

**Summary:**

The reviewers acknowledged the paper's conceptual novelty and clarity in unifying graph learning tasks as inverse problems. However, they raised consistent concerns about: Limited scope and diversity of benchmarks, lack of strong empirical validation, incomplete grounding, and weak evidence of practical utility or cross-domain transfer. Discretization remains a bottleneck without deeper analysis or solutions.

**Reviewer Concerns:**

Addressed concerns:
The authors expanded benchmarks and added supplementary experiments. They clarified the inverse problem formulation and joint training for blind/non-blind cases. They acknowledged missing related work and added references.

Ramianing concerns:
Benchmarks are still considered modest; baselines are not sufficiently competitive or diverse. No significant technical or algorithmic contribution was introduced. Discretization challenges were highlighted but not resolved. Limited evidence that the framework enables cross-method transfer or performance gains.

**Reviewer Scores:**

Reviewer CBpu: Maintained score, due to persistent concerns about scope and baselines.

Reviewer uJ8E: Likely unchanged, as concerns about novelty and benchmark realism were not fully resolved.

Reviewer Ubr3: Likely raise score due to clarifications and additional experiments.

Reviewer JtZo: Likely raise score.

---

### Decision · Program_Chairs · 2026-01-26

Reject